# Structural insights into WRN helicase reveal conformational states and opportunities for MSI-H cancer drug discovery

Catherine T. Fletcher, Abigail A. Mornement ⓘ , Charlotte Barrett, Peter Canning, Prakash Rucktooa, Sophie Huber, Christopher D. O. Cooper, Conor C. G. Scully, Andrew S. Doré, Daniel Rohle, Geoffrey M. T. Smith, Sarah E. Skerratt & Amanda J. Kennedy ⓘ ✉

Werner syndrome helicase (WRN) is a RecQ-family DNA helicase essential for genome maintenance and is a synthetic lethal target in microsatellite instability-high (MSI-H) cancers. Despite its therapeutic promise, the conformational dynamics that enable WRN to unwind DNA, and how inhibitors disrupt this activity, remains poorly understood. Here, we present crystal structures of apo WRN and WRN bound to single-stranded DNA (ssDNA), capturing key conformations in the helicase catalytic cycle. These structures reveal how WRN engages DNA through conserved polar and aromatic interactions, and how domain rearrangements, including an ordering of the aromatic-rich loop (ARL), drive directional translocation. Biochemical and biophysical data demonstrate how nucleotide and inhibitor binding remodel these conformations and suggest that known clinical inhibitors (HRO761 and VVD-133214) function by locking WRN in inactive, 'off-DNA' states. Resistance emerged rapidly in vitro, through acquired point mutations as well as altered WRN expression. Together, our findings provide a structural framework for the WRN structural cycle and support the development of next-generation 'on-DNA' inhibitors to overcome resistance.

Werner syndrome helicase (WRN) is one of the five human members of the RecQ family of ATP-dependent DNA helicases that unwind DNA in a 3′–5′ direction and plays important roles in multiple DNA repair pathways and the maintenance of genome integrity. Biallelic mutations in the human WRN gene cause Werner Syndrome, a rare hereditary premature aging disorder[1], underscoring the essential role of WRN in genome maintenance. Functional genomic screens using CRISPR-Cas9 and RNAi have identified WRN as a synthetic lethal dependency in cells that have high microsatellite instability (MSI-H)[2–5]. Depletion of WRN leads to widespread DNA double-strand breaks, resulting in cell cycle arrest and/or apoptosis in MSI-H cell models but not microsatellite stable cells, and preclinical data supports WRN as a highly promising therapeutic target[6].

WRN is a 1432-residue multi-domain protein comprising a 3′–5′ exonuclease domain, a conserved helicase core, RecQ C-terminal (RQC) domain, and a helicase and RNaseD C-terminal (HRDC) domain. It is the only RecQ family member with intrinsic exonuclease activity, though mutational studies suggest that helicase activity alone is required for survival

of MSI-H cells[4,7]. The WRN helicase core consists of two RecA-like sub-domains (D1 and D2, residues 558–724 and 749–899, respectively) connected by a flexible hinge region. Within the core, several conserved motifs, including Walker A (motif I), Walker B (motif II) and the Aromatic Rich Loop (ARL)[8,9], are implicated in ATP-hydrolysis and DNA translocation[10]. Downstream, the RQC domain (comprising a Zn-binding sub-domain and a winged-helix (WH) domain), and the HRDC domain at the C-terminus, contribute to DNA binding[11,12].

The first WRN helicase-RQC crystal structure[7], reported in 2021, revealed the ADP-binding site at the D1-D2 interface, with key residues from both domains stabilising the adenine and phosphate groups. The adenine base is π-stacked between H546 and K550 and hydrogen-bonded to Q553 of the conserved Q motif. The α- and β-phosphates form polar contacts with D1 residues and the D2 Arginine finger (R857), while K577 interacts with the Walker B motif. A subsequent structure[13] with ATP and Mg$^{2+}$ showed additional contacts between the γ-phosphate and R854, R857, K577, and D668, providing further insight into ATP hydrolysis. While other

CHARM Therapeutics Ltd., B900, Babraham Research Campus, Cambridge, UK. ✉e-mail: amanda@charmtx.com

RecQ helicase structures have been reported[7,10,14–18], key conformations of WRN in its apo and DNA-bound states remain unexplored - limiting our understanding of how its domain dynamics enable DNA unwinding.

In 2023 and 2024, the first WRN inhibitors (HRO761[19] and VVD-133214[13]) entered clinical trials. HRO761 is a potent, selective, non-covalent allosteric inhibitor that locks WRN in an inactive 'twisted' conformation by binding at the hinge region[19]. VVD-133214 is a covalent allosteric inhibitor that targets WRN in the presence of ADP[13]. Despite differing mechanisms, both depend on interactions with C727 within the hinge region, which is covalently modified by VVD-133214 and engages with the triazolopyrimidinone scaffold of HRO761. A third compound, GSK_WRN3[20], also covalently binds C727.

Preclinical data have shown that on-target mutations confer resistance to some WRN inhibitors in the MSI-H colorectal cancer models[21], raising the possibility that clinical resistance may emerge. Should this occur, new strategies to modulate WRN will be needed.

WRN's catalytic function relies on coordinated domain rearrangements that couple ATP turnover to DNA binding and translocation. While previously solved structures have captured WRN in isolated nucleotide-bound states[7,13,19,22], the DNA-bound conformation has remained elusive.

Without it, the full picture of WRN's structural cycle and how inhibitors may exploit or be evaded by these states remains incomplete. Understanding these transitions is critical for the rational design of inhibitors that can either stabilise alternative conformations or overcome resistance mutations.

Here, we report the crystal structures of apo WRN and nucleotide-bound WRN in complex with ssDNA. Together with biochemical and biophysical studies of nucleotide, DNA, and inhibitor binding, our data illuminate the dynamics of DNA unwinding and reveal conformational flexibility essential for WRN function. We contextualize the binding of clinical candidates and present a structural cycle of WRN, providing a mechanistic framework to inform the design of next-generation WRN inhibitors.

## Results
### WRN adopts flexible domain conformations in the absence of DNA
A structure of WRN in its apo form was generated with 1.96 Å resolution (Fig. 1a, PDB 9S1A). In comparison to the previously published nucleotide-bound forms, the overall fold of the D1 and D2 domains is maintained as in the ADP-bound form, but with key conformational changes in nucleotide

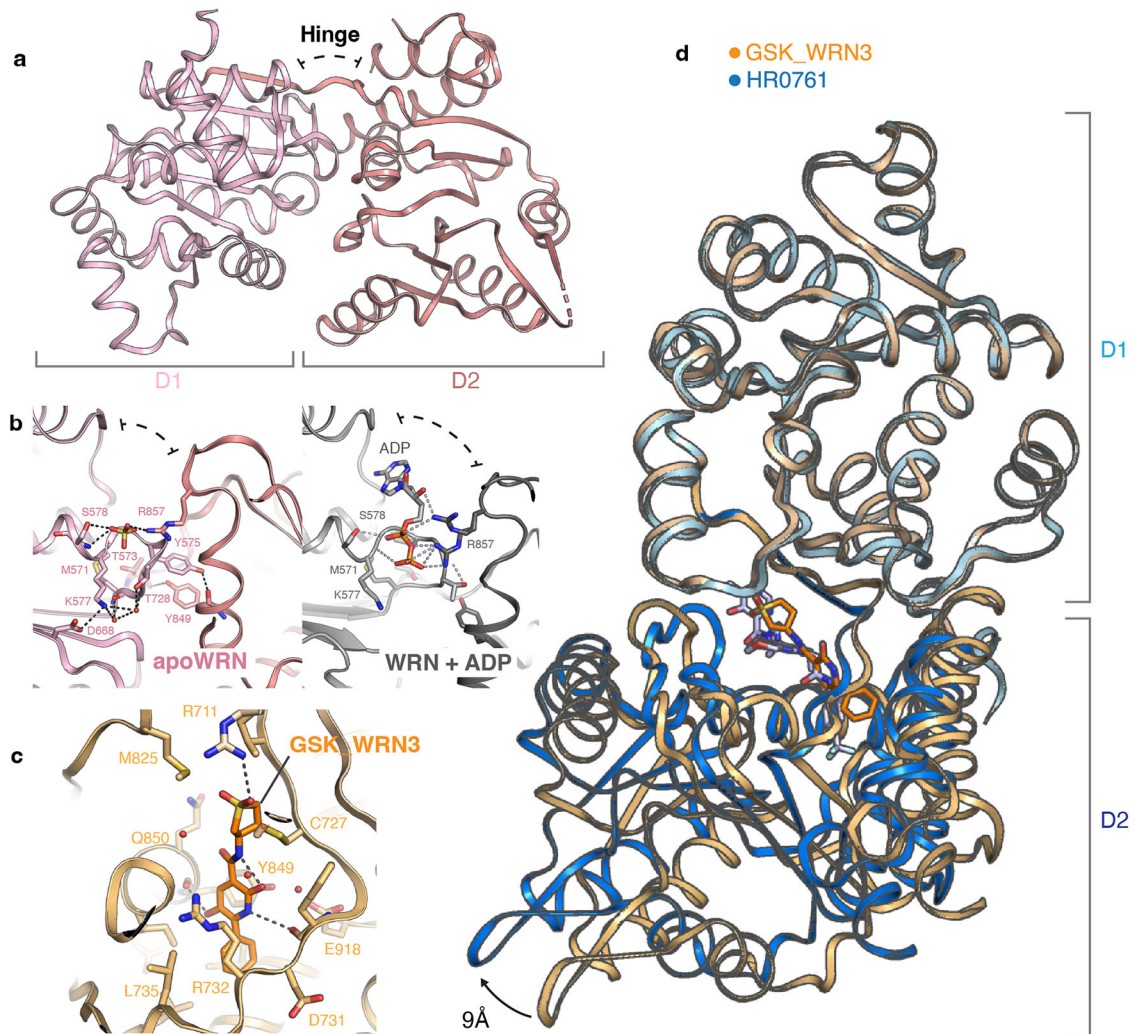

**Fig. 1 | Structural comparison of the WRN in apo and inhibitor-bound conformation. a** Novel structure of apo-WRN (pink) helicase in a 'closed' conformation. **b** The nucleotide binding site reveals a H-bond network stabilises a 'closed' conformation in absence of nucleotide (pink) in comparison to the ADP-bound conformation (grey). **c** GSK_WRN3 binding pose suggests that shape complementarity is sufficient to position the core within the D2 pocket, with a hydrogen bond with between vinyl sulfone and R711 orientating the warhead for covalent modification of C727. **d** WRN in complex with GSK_WRN3 structure (orange) is in the inhibitor-bound 'twisted' conformation; a 180-degree rotation of D2 compared to D1. Overlay with WRN in complex with HRO761 (blue) reveals domain-domain breathing motion is present within the 'twisted' conformation.

binding site (Fig. 1b) and variability in the relative orientation of the D1 and D2 domains. The absence of nucleotide allows residues 570–577 (the Walker A motif) to rearrange within the nucleotide-binding site. K577 forms a polar interaction with D668 (part of the DEAH motif), and a water-mediated interaction with the backbone carbonyl of M571 and side chain of T573; while a buffer sulphate interacts with the backbone amide of S578 and the guanidinium group of R857 from D2. This 'closed' conformation, results in a smaller angle between D1 and D2 in comparison to the previously described structures ($44.2 \pm 0.7°$ in the apo structure, vs $57.0 \pm 5.7°$ in the presence of ATP or derivative, as calculated by Principal Component Analysis). This is facilitated by the formation of a π-π stack between Y575 and Y849, as well as a hydrogen bond between Y849 and the backbone carbonyl of T728. This interaction pulls the D1 and D2 domains closer together, enabled by the flexible hinge region (residues 727–734).

Known WRN inhibitors can capitalise on the inherent flexibility of the hinge region by stabilising an approximate 180° rotation of D2 relative to D1, to form the 'twisted' conformation[19]. Whilst it was initially hypothesised that a large molecule forming extensive and specific interactions, such as HRO761 (PDB 8PFO), would be required to stabilise this conformation, our co-crystallisation trials with the significantly smaller GSK_WRN3 molecule revealed that it can also stabilise WRN in the 'twisted' state (PDB 9S1B, 1.7 Å). This structure unambiguously shows that GSK_WRN3 is covalently linked to C727 (Fig. 1c), consistent with the published information[20]. In contrast to HRO761, which forms extensive interactions with both helicase domains of WRN, GSK_WRN3 forms limited polar interactions: between the ligand vinyl sulfone moiety and R711 (from D1), and with the backbone carbonyl of residues F730 and Y849 (within D2). Shape complementarity between the core of the molecule and the pocket, involving Van der Waals interactions, is sufficient to lock the protein in this 'twisted' conformation.

Our internally generated crystal structure of HRO761 in complex with WRN[541-943] superposes well with the previously published structure (RMSD 0.465 Å, PDB 9S18, 2.22 Å), displaying limited variability in the D1-D2 domain rotation, despite being generated using different constructs, under different crystallisation conditions and resulting in a different unit cell. In contrast to the structure of WRN in complex with HRO761, the structure in complex with GSK_WRN3 shows a maximal 9Å displacement of D2 relative to D1, despite growing in the same buffer conditions and forming crystals in the same space group (Fig. 1d). Two further structures of WRN in complex with inhibitors targeting this inactive conformation, one of which covalently targets C727, (PDB 9OG3 and 9OG8)[23] show an even greater maximal displacement of 16 Å in comparison to the HRO761 structure, indicating that motion is not only caused by the covalent bond at C727. The ability of WRN to adopt these twisted conformations indicates flexibility in the relative orientation of the D1 and D2 domains.

## WRN engages ssDNA in both apo and nucleotide-bound states

To assess if the presence of nucleotides affects the ability of WRN to bind DNA, a Surface Plasmon Resonance (SPR) assay was established. For the purposes of this work, a ssDNA substrate was used as a surrogate to the more complex DNA structures that have unpaired regions, such as flap-strands or replication fork intermediates, to which WRN is known to bind[24,25]. WRN binding to ssDNA was assessed in the presence or in the absence of nucleotides. Interestingly, it was found that WRN could bind to ssDNA both in the absence, and presence of 1 mM non-hydrolysable ATP-analogue, ATPγS (Fig. 2a, 2 μM and Fig. 2b, 1 μM, respectively), but with different binding kinetics. In the absence of ATPγS, WRN could bind DNA with fast on- and off-rates ($1 \times 10^5 \, M^{-1}s^{-1}$ and $2 \times 10^{-1} \, s^{-1}$). In the presence of ATPγS, WRN was found to bind to DNA with slower on- and off-rates ($4.4 \times 10^4 \, M^{-1}s^{-1}$ and $4.4 \times 10^{-2} \, s^{-1}$), albeit with a similar affinity. A lower affinity binding event was seen with 1 mM ATP (Fig. 2c, $K_D = 18$ μM) and a further drop in binding affinity was seen with WRN + 5 mM ADP, where the $K_D$ was unmeasurable under the concentration range tested (Fig. 2d, $K_D > 400$ μM). Since WRN can bind to DNA in the presence and absence of nucleotides, it was important to confirm that ATP could still bind to the WRN + DNA complex. Further SPR experiments showed that ATPγS

could bind to WRN alone with a $K_D$ of 6 μM (Fig. 2e), and similarly ATPγS could bind the WRN-ssDNA complex with a $K_D = 5$ μM with slower on/off rates (Fig. 2f). This data suggests that WRN, ATPγS and ssDNA can form tri-complexes independent of the binding order. Further competition experiments using fluorescence polarisation (FP) showed that ADP binds to WRN alone with a similar affinity to ATP (Supplementary Fig. 1).

## DNA-bound WRN reveals a domain 'breathing' motion linked to helicase activity

To further enrich understanding of how WRN can facilitate DNA unwinding, we report the structure of WRN in complex with ATPγS and a ssDNA fragment (PDB 9S19, 2.3 Å). Although $Mg^{2+}$ was included in the buffer system, no density corresponding to the ion was observed, with the gamma-sulphate of ATPγS forming interactions with D669 and S578, where the $Mg^{2+}$ was anticipated to bind (Fig. 3a). Two molecules were observed in the asymmetric unit; while the monomers show a high degree of structural homology (RMSD = 0.532), density for more ssDNA bases at the 5′-end could be modelled in chain B, possibly due to its ability to pack against the symmetry-related molecule. Given that WRN shows structural specificity as opposed to rigid sequence specificity[26], it is likely that there is variability in the bound portion of the ssDNA within the crystal preventing absolute assignment of the ssDNA sequence, nevertheless the phospho-deoxyribose backbone was unambiguously modelled.

The overall fold of the individual RecQ helicase domains is maintained in comparison to the ATP-bound structure (PDB 7GQT[13]) (Fig. 3b), with ssDNA binding into a positively-charged groove between the D2 and zinc-binding subdomain (Supplementary Fig. 2A). Polar interactions are formed between highly conserved D2 residues R774 (motif IV), T821 (motif V) (Fig. 3c, and Supplementary Fig. 2B) and the phosphate backbone, as could be expected for a non-sequence specific helicase. An inward shift of helices (residues 870–904) allow the intercalation of L879 between adjacent bases (Fig. 3c), as well as the hydrogen bond between conserved K843 and the nucleotide base of the ssDNA (Fig. 3d), which may act as a ratchet to prevent backsliding. A composite image, combining this structure of WRN helicase core in complex with ssDNA with the RQC domain of WRN in complex with double-stranded DNA (dsDNA) (PDB 3AAF[12]), closely overlays with the previously published structure of another RecQ helicase, BLM + DNA (PDB 4O3M[18]), suggesting a conserved DNA binding groove (Supplementary Fig. 2C, D).

Upon binding of ssDNA, the highly conserved aromatic loop (ARL) transitions from an unstructured loop to an α-helix (Fig. 3d). The aromatic side chain of F680 is intercalated between the ssDNA bases, forming a π-stacking interaction, a structurally conserved motif observed in BLM[27]. The conformation is stabilised by additional polar interactions between the ARL and D1/D2 residues (R681 to E644), and an edge-face π-stack against W676 in chain A. The structuring of the ARL causes a shift of DEAH residues within the nucleotide binding site, bringing them into closer proximity with ATPγS (Fig. 3a, ~1.5 Å). This increased proximity may be responsible for the increased rate of ATP-turnover in the presence of DNA (Supplementary Fig. 3A, B).

## Allosteric inhibition disrupts WRN conformational dynamics and DNA binding

To interrogate further the degree of WRN domain flexibility required to bind DNA, we investigated whether DNA could bind in the presence of HRO671 and GSK_WRN3 (Fig. 4a). Using the SPR assay previously described, with DNA captured on the chip, WRN was pre-incubated with either HRO761 or GSK_WRN3 at 100 μM concentration, in the presence of 1 mM ATPγS. WRN was no longer able to bind to DNA, suggesting that the relative rotation of the D2 to D1 domains results in a conformation of WRN that is not compatible with DNA binding. While the ARL is still accessible in these structures, ssDNA would only be able to interact with either the ARL or the D2 domain due to the rotation of the D2 relative to D1. In longer constructs (encompassing the helicase, RQC and WH domains), binding is still maintained via the positively charged groove between the RQC, $Zn^{2+}$-

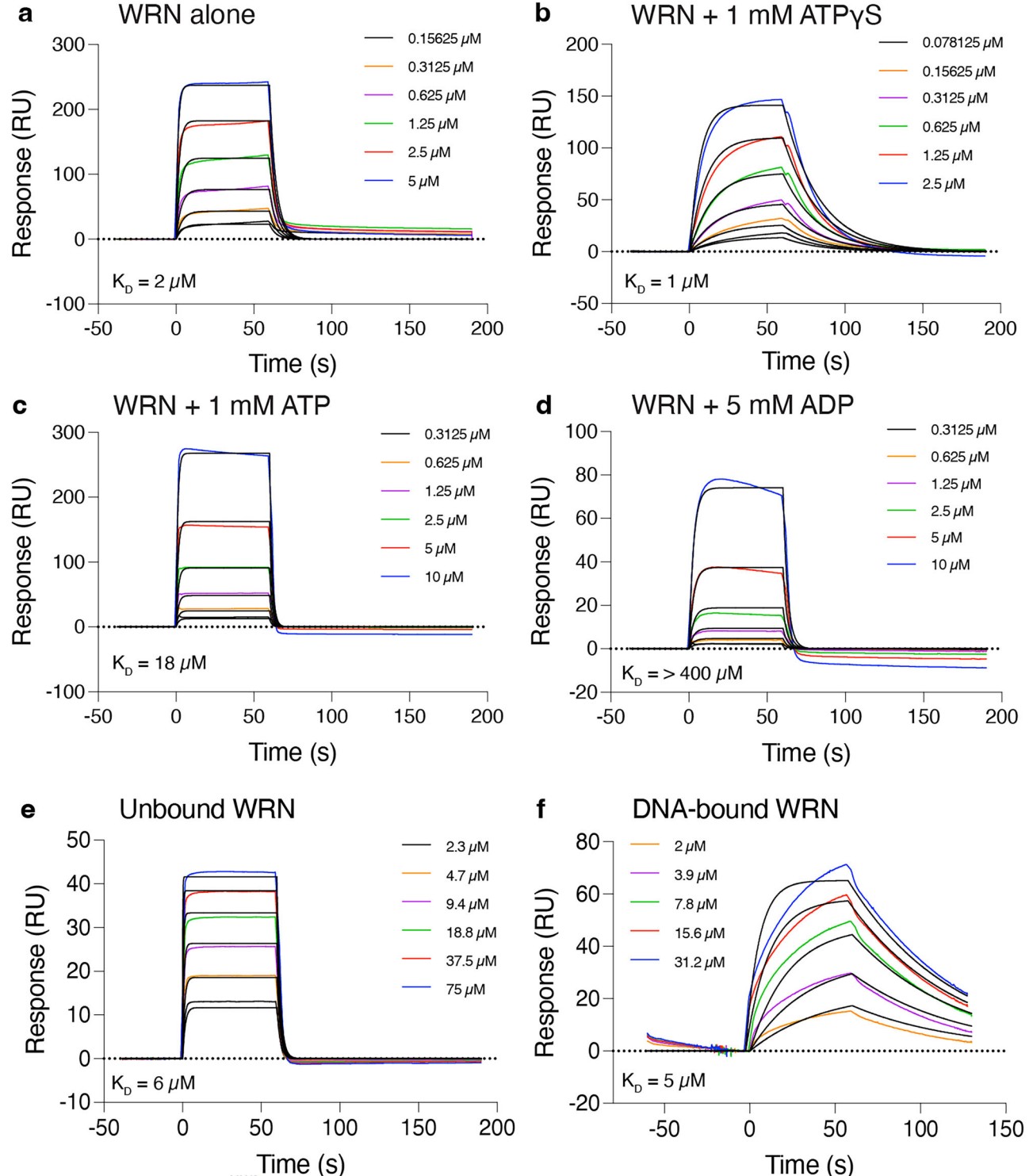

**Fig. 2 | Biophysical characterisation of WRN binding to DNA. a** SPR data measuring the binding kinetics of **a** WRN to ssDNA alone and in the presence of **b** ATPγS, **c** ATP and **d** ADP. SPR data measuring the binding of ATPγS to WRN in the **e** free and **f** DNA-bound states. In all cases the data were fitted to a 1:1 kinetic binding model (black line).

binding and D2 domains, although unwinding is inhibited through the displacement of the ARL, as previously reported for HRO761[19].

At the time of this work, the clinical compound from Vividian Therapeutics (VVD) and its binding mode, beyond the covalent targeting of C727, had not been disclosed. Competition assays with the ATPγS FP probe demonstrated that despite VVD-133214 and GSK_WRN3 both targeting C727, VVD-133214 binds cooperatively in the presence of ATPγS (Fig. 4b),

in agreement with the literature[13], while HRO761 and GSK_WRN3 exhibit overlapping binding with the nucleotide binding site. Co-crystallisation of patent exemplar molecule **81**[28] in the absence of nucleotide closely recapitulates the published structure of VVD-133214 solved in the presence of ADP (Fig. 4c, PDB 9S17 and 7GQS[13] respectively), with conserved motion of Phe917 to accommodate the core and covalent modification of C727. Notably, the overall fold of the WRN + VVD compounds, in both presence

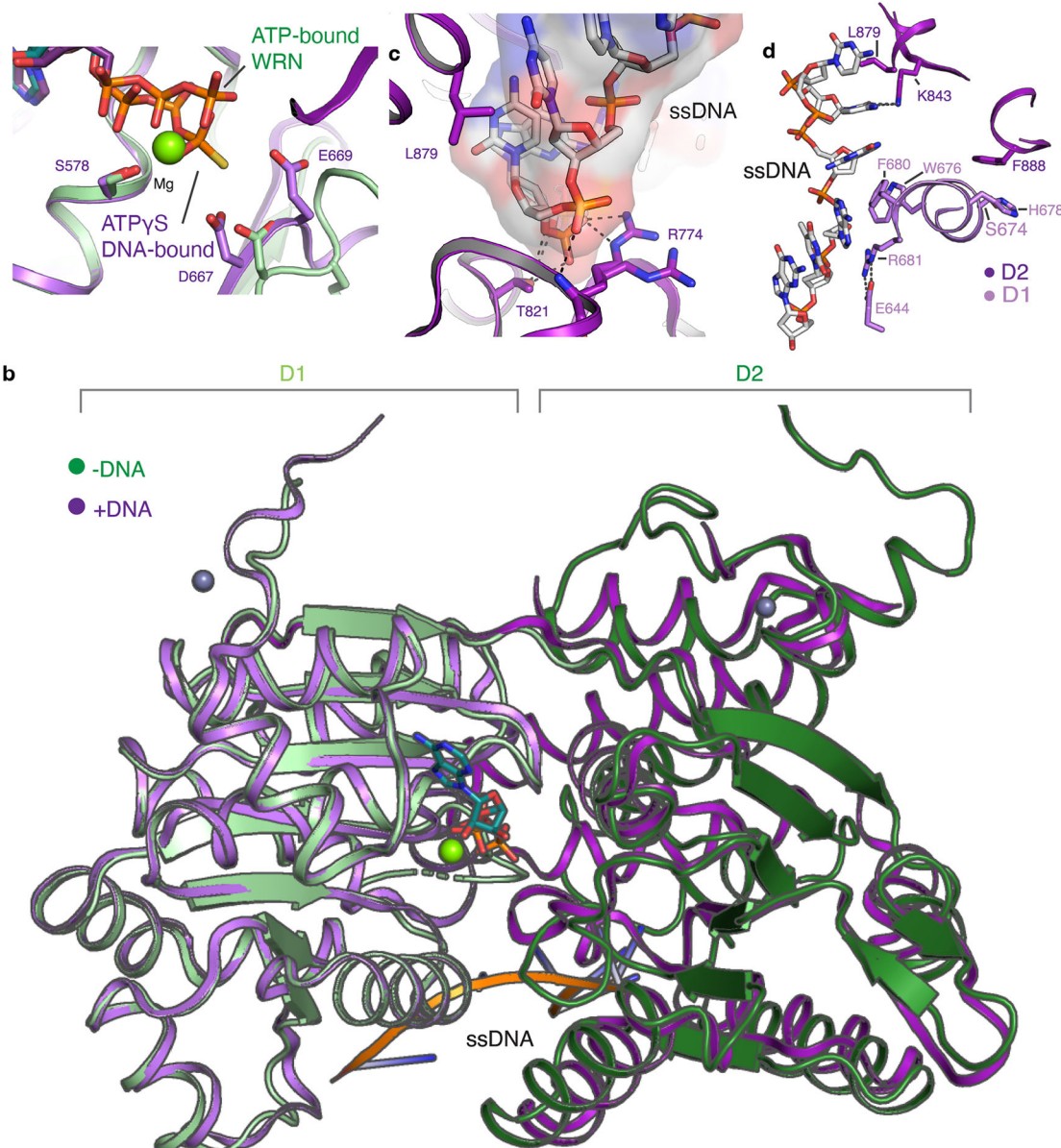

**Fig. 3 | Structural characterisation of WRN binding to DNA. a** Overlay of the nucleotide binding sites of WRN + ATP (PDB 7GQT, green) with WRN in complex with ATPγS and ssDNA (purple), showing absence of bound-$Mg^{2+}$ ion and the contraction of the DEAH motif in the presence of ssDNA and ATPγS. **b** Super-imposed structures of ATP-bound WRN (green) and WRN in complex with ATPγS and ssDNA (purple). **c** Interactions between the D2 domain and ssDNA, high-lighting the conserved ratchet mechanism. **d** ARL forms α-helix upon binding of ssDNA, interchelating the side chain of F680 between DNA bases, and stabilised by polar interactions (black).

and absence of nucleotide, shows a high degree of structural homology with the structure of apo WRN (RMSD 0.487 Å between the two ligand-bound structures, and 1.058 Å between 7GQU and the apo conformation). The ligand-bound structures show stabilisation of the Walker A motif across the nucleotide-binding site, and a water-mediated hydrogen bonding network between D668, K577, T573 and the backbone carbonyl of M571, as well as the π-π stack between Y575 and Y849, a similar 'closed' conformation as apo WRN. Further, the β-phosphate of ADP in the VVD-133214 structure occupies the same space as the buffer sulphate in the apo and molecule **81** bound structures, forming the same interactions with S578 and R857 (Fig. 4d). Although VVD-133214 binds to an alternative binding pocket to HRO761 and GSK_WRN3, in our SPR assay it similarly prevents binding of WRN to ssDNA (Fig. 4a). Superposition of the DNA + ATPγS-bound structure of WRN with the molecule **81**-bound structure shows clear steric clash between the molecule **81** and the helix containing motif III (residues

700–707), indicating that this molecule traps WRN in the 'closed' con-formation, preventing the opening of D1 and D2 domains, to expose D2 residues required for full DNA binding (Supplementary Fig. 4). Overall, this suggests that inhibition of conformational changes through the hinge region impedes binding of WRN to DNA.

**Clinical WRN inhibitors select for on-target resistance mutations**
HRO761 and VVD-133214 are potent WRN inhibitors (Supplementary Table 1) that lead to increased dsDNA breaks and cell death selectively in MSI-H cell lines. In addition to halting unwinding activity by locking WRN in conformational states that prevent ssDNA binding; using a HiBit-tagged WRN cell line to monitor protein turnover, both inhibitors induced degradation of the WRN protein ($DC_{50}$ = HRO761: 270 nM and VVD-133214: 145 nM) (Supplementary Table 1). Chronic treatment (Supple-mentary Fig. 5) of HCT116 cells with either HRO761 or VVD-133214 was

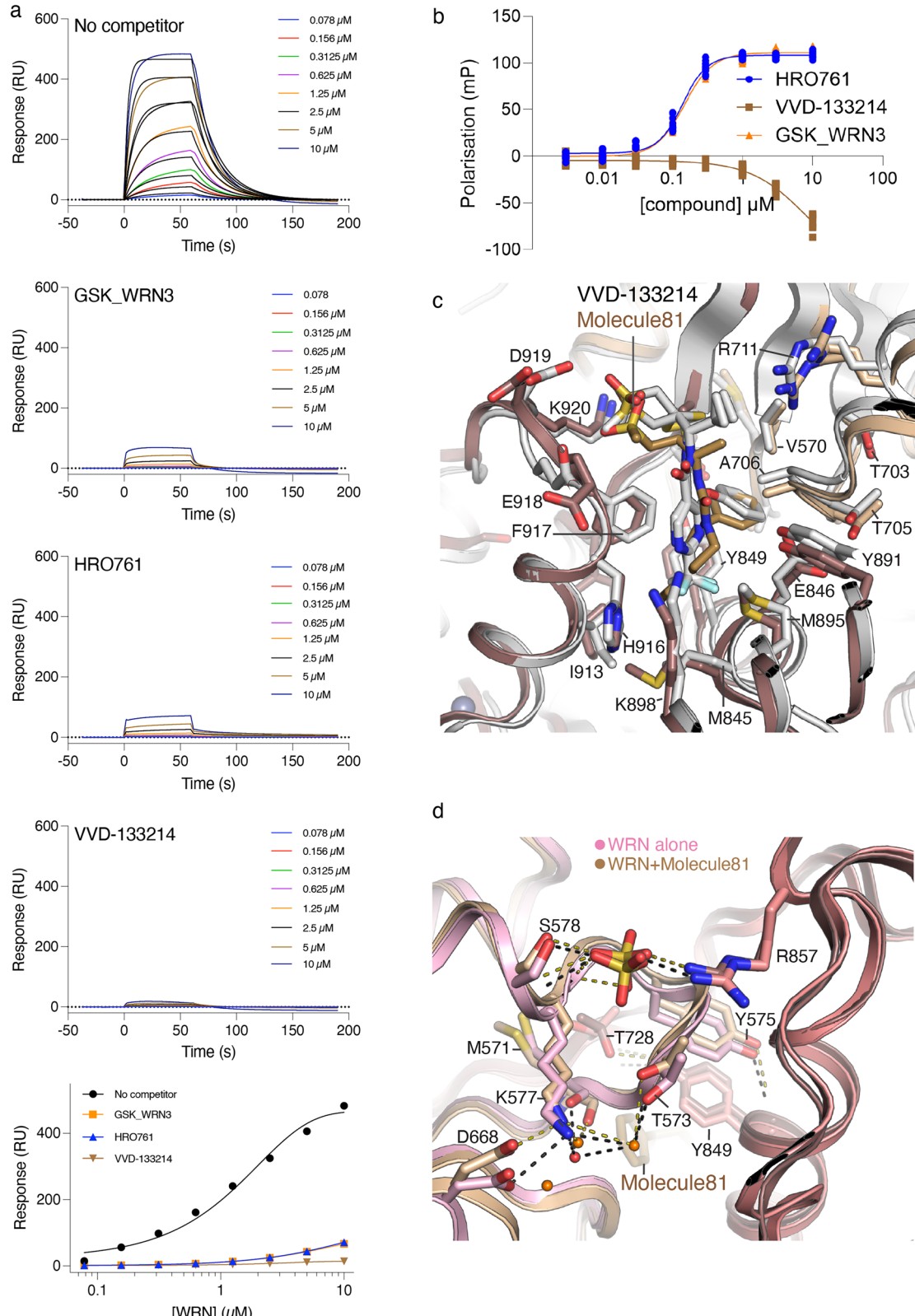

**Fig. 4 | Two independent mechanisms of inhibition prevent WRN binding ssDNA.** ssDNA was immobilised on a chip and binding of: WRN + ATPγS; or WRN + ATPγS in the presence of GSK_WRN3, HRO761 or VVD Example 81 was measured using SPR. **a** Summary plot of the kinetic response of each condition shows that all inhibitors block WRN binding to ssDNA. **b** Using a bodipy-ATPγS competition assay, in the absence of DNA, inhibitor concentration response curves show differentiated mechanisms of inhibition for VVD-133214 (yellow) compared to HRO761 (blue) and GSK_WRN3 (orange). Data shown as individual data points from $n$ = 3 technical replications. **c** Overlay of WRN in complex with molecule 81 + no nucleotide (yellow), with WRN + VVD-133214 + ADP (PDB 7GQU, grey), shows a consistent protein fold at the inhibitor binding site independent of nucleotide presence. **d** Overlay of WRN in complex with molecule 81 + no nucleotide (yellow) with the apo structure (pink) shows a conserved fold in the nucleotide binding site and a conserved H-bond network.

carried out to interrogate potential resistance mechanisms that could arise from this on-target activity, and we present the data with the structural context of WRN inhibition. Analysis was carried out on samples around day 50, which represents early resistance mechanisms, and later after chronic treatment with the highest doses (HRO761: 13 μM, ~120 days and VVD-133214: 3 μM, ~60 days).

The HCT116 HRO761-treated cell line showed >500-fold resistance to further treatment (untreated: HRO761 $IC_{50} = 76$ nM; treated: HRO761 $IC_{50} > 30,000$ nM) and exhibited modest cross resistance to VVD-133214 (untreated: VVD-133214 $IC_{50} = 6$ nM; treated: VVD-133214 $IC_{50} = 135$ nM) (Fig. 5a, b). RNASeq data from a pooled source of cells, revealed a clear signal in the variant analysis for a single on-target point mutation, F730S, with an average allele frequency of 93% across three biological replicates. While F730 forms the edge of the HRO761 binding site, it forms no direct interactions with the ligand, instead sitting directly downstream of hinge residue, Gly729, that facilitates the rotation between the D1 and D2 domains (Fig. 5c). The sidechain of Phe favourably occupies different hydrophobic pockets in each of the apo and ligand bound structures, and so substituting Phe with Ser will disrupt these hydrophobic contacts which could explain the marked loss in compound potency.

Neither this residue, nor the 'twisted' conformation, are involved in the binding of VVD-133214 (Fig. 5d) and so this point mutation is not anticipated to impact the activity of VVD-133214. Further analysis of the differentiated gene expression showed that the most noticeable change in pathway dynamics was upregulation of WRN expression (Fig. 5e) which increased with passages of treatment. This could explain the drop in potency of the highly potent molecule VVD-133214 in the HRO761-treated cell line. This significant upregulation in WRN protein expression was confirmed by western blot analysis (Fig. 5f).

The VVD-133214-treated cell lines showed >300-fold resistance to further treatment (untreated: VVD-133214 $IC_{50} = 5$ nM; treated: VVD-133214 $IC_{50} = 1881$ nM), whilst remaining sensitive to HRO761 (untreated: HRO761 $IC_{50} = 95$ nM; treated: HRO761 $IC_{50} = 132$ nM) (Fig. 5g, h). Variant analysis of the RNASeq data, from a pooled sample of this cell line, also revealed a single point mutation, T705A with an average allele frequency of 87% across three biological replicates. Interestingly, this residue was first mutated to Ile, at an average allele frequency of 56%, at the earlier stages of resistance generation, suggesting that decreasing the size of this residue leads to progressive reduction in potency of the VVD molecule. In both the apo- and the VVD-bound conformations (both published and internally generated), this Thr residue is involved in the H-bond network with E846 and S707 that is not present in the ATPγS-bound form in either the presence of absence of ssDNA (Fig. 5d). A mutation to a non-polar residue such as Ile or Ala could partially disrupt this network and lead to reduced compound effect. While upregulation of WRN was also observed (Fig. 5f), this only occurred in the late stage of resistance and so does not solely explain the drop in potency as seen in the early passages.

## Discussion

The apo- and DNA-bound WRN crystal structures presented here provide critical insights into the conformational transitions that govern WRN's interaction with DNA - transitions that are essential to both its function and pharmacological modulation. In Fig. 6a, we illustrate the structural cycle of WRN translocation following analysis of both these and previously published conformations of WRN. Structural characterisation has demonstrated variation in the orientation between the RecQ domains, D1 and D2, propagated through the hinge region, which we refer to as a 'breathing motion'. We refer to the 'closed' conformation as the state where the angle between D1 and D2 is smallest; and 'open' conformation where the angle is increased. The ability for WRN to bind DNA has more complexity than angle between D1 and D2 alone; and is connected to the specific movements in the nucleotide binding site and D2 domain for which we now have structural insight and hence will refer additionally to 'on-DNA' and 'off-DNA' conformations independently.

In apo state, WRN adopts a 'closed' conformation, resulting in greater proximity between the two domains at the hinge region (Supplementary Fig. 6 and Supplementary Table 2). Binding of ATPγS with or without ssDNA, induces an opening motion, increasing the angle between the two domains and increasing the separation of equivalent residues at the DNA interface. DNA itself binds to a groove within WRN, stabilised by a hydrogen bonding pattern between a conserved lysine and the nucleotide base of ssDNA as well as intercalation of non-polar residues from D2 between DNA bases. Together, this forms a ratchet to enable unwinding in one direction[29]. The increase in D1-D2 separation may serve to 'push' WRN along DNA, to promote strand separation.

Additionally, when DNA is bound to WRN in the presence of ATP, there is a conformational change in the ARL, with the aromatic side chain of F680 intercalating between ssDNA bases. The transition from unstructured to structured ARL, and contraction of the nucleotide site to promote ATP hydrolysis, reflects the broader flexibility of WRN and leads to unwinding activity. This mechanism is observed in BLM helicase[18] and is consistent with mutational studies of residues within the ARL in other RecQ[10,14] and related SF2 helicases[15] that suggest there is allosteric communication between the ARL and nucleotide binding site.

Following hydrolysis, the instability of the WRN + ADP + ssDNA complex, as indicated by WRN + ADP exhibiting a reduced affinity for ssDNA compared to WRN + ATP or WRN alone, likely results in ADP release. Nucleotide-free WRN would then undergo further conformational changes: the D1 domain is released from ssDNA, with the ARL reverting to the unstructured conformation and no longer intercalating the conserved F680 between ssDNA bases. The orientation between D1 and D2 reverts to the 'closed' fold, where D1 is in closer proximity to D2 at the DNA interface. The D2 and RQC domain remain stabilised on the DNA, through extensive polar interactions and engagement of the 'ratchet' to prevent backsliding, as observed in the published structures of other RecQ helicases in complex with DNA (PDB 2WWY[30] and 4TMU[16]). Direct overlays of apo, ATPgS +ssDNA-bound and ADP-bound WRN (PDB 6YHR[7]) suggest that when ADP is bound (in comparison to no nucleotide or ATPgS) there is an outward swing of the helices formed by residues 870–904. This movement may limit engagement of the ratchet and contribute to the reduced affinity for DNA observed, although further work would be required to explore this. This opening and subsequent closing shifts the whole helicase domain along the DNA. ATP can now occupy the empty nucleotide binding site, enabling re-engagement of D1 and ARL with the ssDNA, restoring the coordinated ssDNA interactions across the ARL and D2 surface required for productive helicase activity, and the cycle repeats.

While known inhibitors, HRO761 and VVD-133214, differ in their binding mode (non-covalent vs. covalent, respectively) both converge on C727, either leveraging structural positioning or forming direct covalent bonds to stabilise inactive states. Mechanistically they lock WRN in a specific state, blocking this 'breathing' motion to inhibit helicase activity (Fig. 6b). Whilst our internal data indicates a loss of DNA binding to the WRN helicase core in the presence of inhibitors, published data utilising a longer construct including the RQC domain (WRN[519-1227] or WRN[519-1238]) demonstrates the non-competitive nature of the HRO761 and VVD-133214 with respect to DNA[13,19]. Inclusion of the RQC domain and use of dsDNA likely enables binding of WRN to dsDNA via the RQC and the conserved positive groove found on the D2 helicase domain. HRO761 and GSK_WRN3 lock WRN in the 'twisted' confirmation that disconnects the RQC and D2 domains from D1 while VVD-133214 and related molecule **81** appear to lock WRN in the 'closed' conformation. Both mechanisms would prevent engagement of the ARL on the ssDNA, which in turn would disconnect binding of DNA to nucleotide hydrolysis, resulting in reduced activity, as previously shown for other RecQ and SF2 helicases[10,15]. Binding to dsDNA may still be possible under certain conditions, such as with inclusion of the WH domain. Our study is limited to using a helicase-only construct and ssDNA, but the alignment of composite structures including ssDNA and dsDNA, with close analogue BLM helicase in the presence of DNA shows concordance of these findings. This suggests that our results

**Fig. 5 | Known WRN inhibitors develop acquired resistance through single point mutations.** HCT116 cells were chronically treated with HRO761 for ~120 days and inhibitor response of **a** HRO761 and **b** VVD-133214 on cell proliferation were measured. **c** Overlay of crystal structures of apo-WRN (pink) and HRO761-bound WRN (blue) show that F730 is on the hinge region between the two domains next to the HRO761 binding site. **d** The WRN + molecule **81** crystal structure (yellow) shows that F730 is not directly involved with inhibitor binding; whereas T705 is involved in the H-bond network with E846 and the backbone of S707, to stabilises the 'closed' conformation of WRN when the inhibitor is bound. **e** RNASeq analysis of gene expression in the acute- and chronically treated resistant cell lines compared to controls (DMSO-treated and parental); *$p$adj < 0.001. **f** Elevated WRN expression levels were validated through western blot analysis of cell lysates. HCT116 cells were chronically treated with VVD-133214 for ~60 days and inhibitor response of **g** HRO761 and **h** VVD-133214 on cell proliferation was measured. Cellular data in graphs is shown as individual data points from *n* = 3 technical replicates.

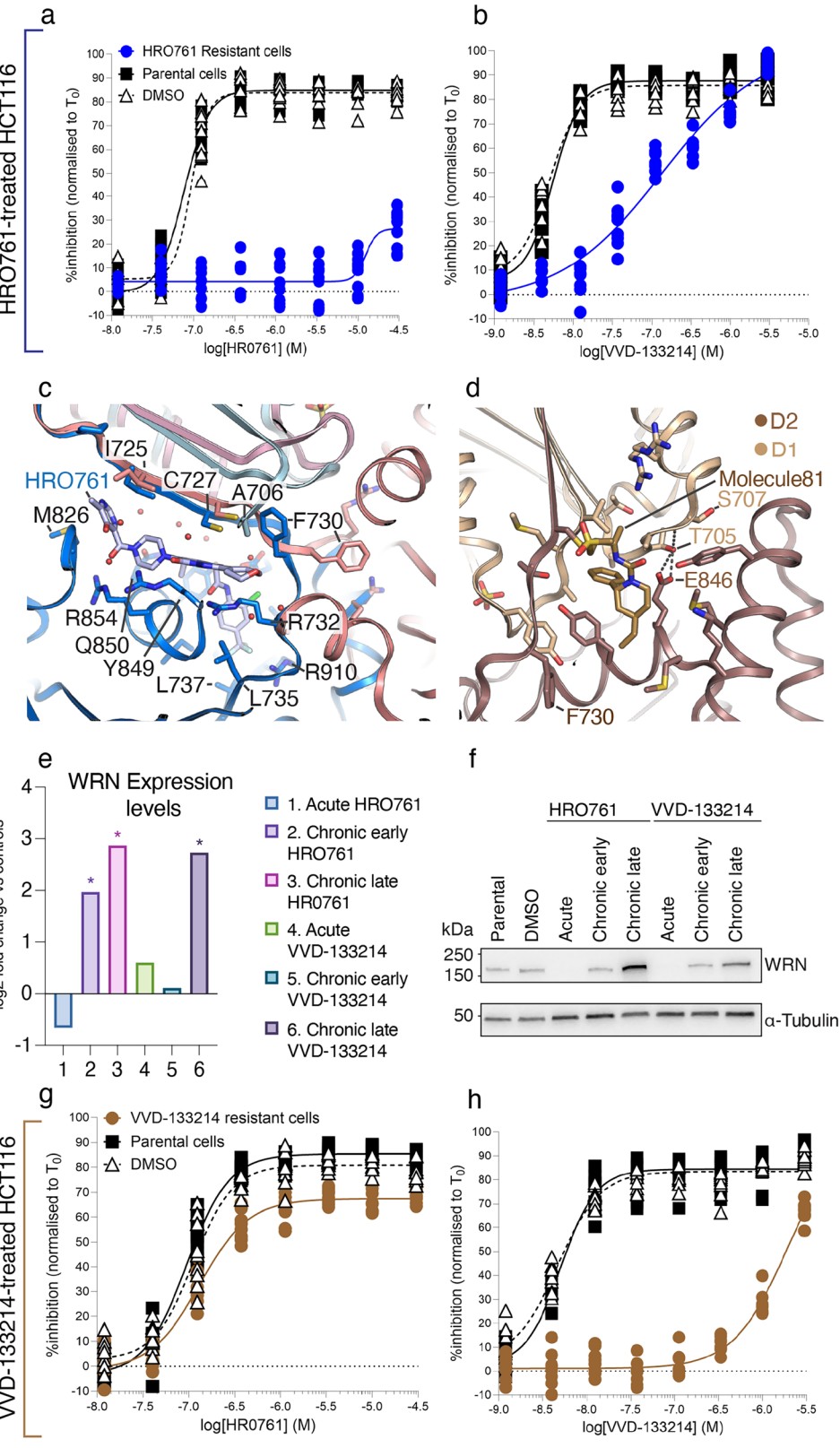

have relevance to the physiological setting of full-length WRN binding to DNA structures in cells.

As WRN inhibitors advance clinically, identifying the molecular adaptations that cause resistance will be key to developing more effective and sustainable treatment approaches. Inhibition of WRN in MSI-H cells leads to activation of the DNA damage response pathways, which activates ATR and/or ATM kinases, leading to ubiquitination of WRN and subsequent degradation[19,31]. Chronic treatment of MSI-H cells with either HRO761 or VVD-133214 rapidly led to the development of resistance. This has been attributed to increased WRN turnover driving compensatory protein synthesis, as well as the induction of senescence which may provide time for adaptive changes[32].

**Fig. 6 | The WRN lifecycle and opportunities for alternative mechanisms of inhibition. a** Structural insights into the dynamics of WRN helicase, zooming in to look specifically at the nucleotide binding site (upper left) and the DNA binding ARL region (lower right) of the different conformational states of WRN "off-DNA" (yellow) and "on-DNA" (green), and the transitions between. ATP is highlighted in magenta, ADP in yellow, and the ARL is in salmon, highlighting the transition from unstructured to structured α-helix. PDB codes of structures shown: ssDNA only (4TMU, from related RecQ helicase), ATP+ssDNA (PDB 9S19), Apo (PDB 9S1A), ADP (PDB 6YHR), ATP (7GQT). **b** Graphical summary of WRN helicase. When not bound to DNA, WRN "off DNA" exhibits a breathing motion (yellow) dependent on nucleotide binding, where the linker region gives flexibility for the domains to move relative to each other. These conformations are inhibited by two distinct mechanisms - the first leads to a 180-degree rotation of D1 in relation to D2, putting WRN into the "twisted" conformation, whilst the other blocks this breathing motion by halting WRN in a "closed" conformation. When bound to DNA the linker region allows the domains to separate further (green) to accommodate DNA binding, and as yet the conformation WRN "on DNA" is undrugged and could provide additional opportunities for drug development.

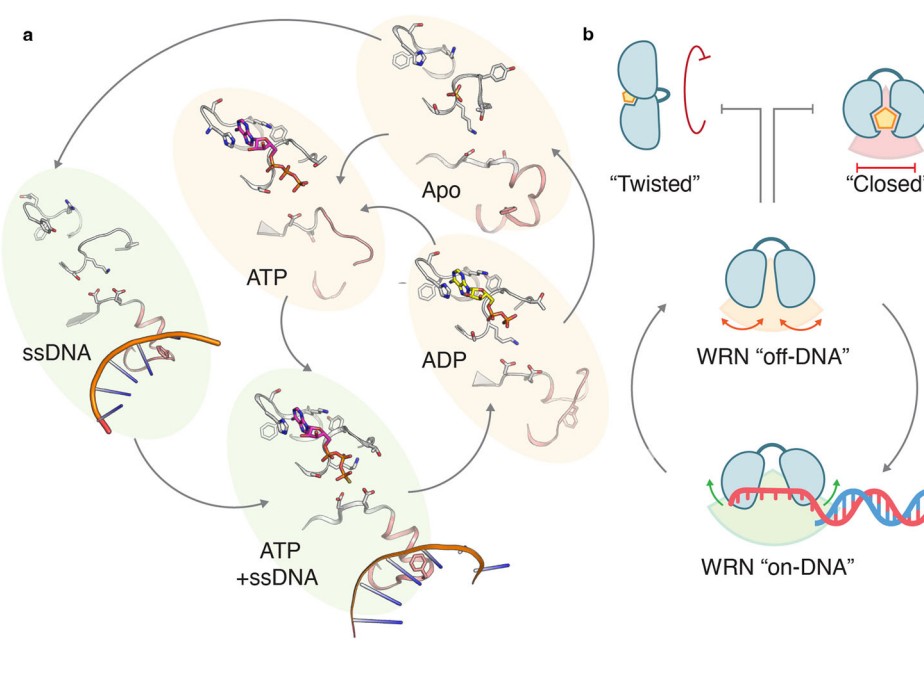

Bulk RNASeq analysis, despite limitations in sensitivity due to being from a pooled source of cells, revealed striking evidence of single point mutations, with high allele frequency, in the transcriptome of HCT116 cells treated with either VVD-133214 or HRO761. The F730S mutation identified following treatment with HRO761 agrees with work carried out by Picco et al.[21], who identified both G729 and F730 as point mutations following treatment with either HRO761 or GSK_WRN3. In contrast, chronic treatment with VVD-133214 induced a single point mutation at position T705. In addition to these high-frequency alterations, WRN gene expression was significantly upregulated (>2-fold in late passages), which could be a further source of resistance. This expression upregulation, seen in other kinase-driven cancers (e.g. BCR-ABL[33] or ALK[34]) may blunt the potency of otherwise efficacious compounds like VVD-133214. These findings suggest at least two primary resistance pathways: (1) mutations that alter inhibitor binding at the hinge region and (2) compensatory upregulation of WRN expression.

While biochemical validation of resistance mutants lies beyond the scope of this study, our findings lay important groundwork. Future work to characterise the binding and functional activity of mutant WRN proteins will be necessary to fully evaluate their impact on drug efficacy. Nevertheless, our data demonstrate that high-frequency binding site mutations can profoundly alter the activity of current WRN inhibitors, supporting the need for new strategies to effectively drug this target. This echoes the BCR–ABL resistance story, where first-generation ATP-site inhibitors like imatinib were compromised by mutations such as T315I[35] but, allosteric inhibitors like ABL001[36], provided renewed clinical benefit[37].

Since completing this work, a new clinical candidate, GSK4418959 (IDE275), of which GSK_WRN3 is a precursor molecule, has been disclosed[38]. Preliminary data suggest that GSK4418959 engages WRN in a conformationally distinct state relative to HRO761 and VVD-133214, potentially enabling it to overcome certain resistance mutations. Although the available data imply a related 'twisted' conformation, structural details remain limited and were presented only after the conclusion of our study. As such, GSK4418959 is not included in our analysis. Future studies comparing its binding mode

with the conformational states presented here will be important to define its mechanism of action and therapeutic potential.

By disclosing the crystal structures of apo WRN and WRN bound to ssDNA, we present a more complete picture of the conformational states adopted by WRN throughout its functional cycle. This structural visualisation highlights the requirement for domain flexibility to enable productive DNA engagement with the ARL and unwinding. With current clinical molecules shown to selectively bind the 'off-DNA' state, and with resistance mechanisms now emerging, we propose that the 'on-DNA' conformation offers a compelling opportunity for the development of second-generation inhibitors, which could potentially enable a 'trapping' mechanism similar to that exploited by PARP inhibitors[39]. In the specific context of overcoming on-target resistance, such inhibitors may exhibit orthogonal binding profiles, with potential to improve patient outcomes both as monotherapies and in combination with existing 'off-DNA' inhibitors in MSI-H cancers.

## Methods
### Materials
WRN inhibitors, HRO761 (HY-148699), VVD-133214 (HY-158116) and GSK_WRN3 (HY-162471) and Staurosporine (HY-15141) were available to purchase from MedChemExpress. Patent exemplar, molecule 81, was synthesised following the procedure as outlined in Kikuchi et al. Patent Application US 2024/0140915 A1[28].

### Protein production
All constructs, comprising the helicase domain of hWRN (UniProt ID Q14191 either 517–941 or 517–943) with an N-terminal hexa-histidine tag and TEV protease cleavage site, were cloned into a pOET1.1 vector (Oxford Expression Technologies) for recombinant baculovirus generation. Proteins were expressed in Spodoptera frugiperda (Sf21) cells at 27 °C under constant shaking for 48 h. Cells were harvested by centrifugation and pellets stored at −80 °C.

The different constructs of WRN required optimised buffer conditions for purification (full details in supplementary methods). Briefly, the cell pellet was resuspended in a lysis buffer supplemented with protease

inhibitors and nuclease. Cells were lysed using a high-pressure homogenizer system and the lysate was cleared by centrifugation for 45 min at $30,000 \times g$. Clarified lysate was incubated with HisPur Ni-NTA resin (ThermoFisher) pre-equilibrated in a lysis buffer at 4 °C for 1.5 h. Resin was washed with a high salt buffer, followed by a wash buffer, prior to stepwise protein elution. Elution fractions containing protein were pooled together, and N-terminal tags were removed by adding TEV-protease overnight. TEV-protease was then removed by re-applying the sample onto Ni-NTA resin. Fractions containing protease-free protein were pooled and concentrated before applying it onto a HiLoad Superdex 200 pg column, pre-equilibrated with SEC buffer. Protein-containing fractions were pooled and concentrated to 30 mg/ml, flash-frozen and stored at −80 °C.

### Protein crystallization, data collection and structure solution
Sitting drop, vapour diffusion crystallisation was done using SwissCi MRC 3drop plates on a Mosquito pipetting robot (SPTLabtech), with varying drop ratios, and with WRN typically at 10 mg/mL. Crystallisation plates were incubated and monitored for crystal growth at 20 °C, except for WRN in complex with HRO761, which were incubated at 4 °C and GSK_WRN3, which was grown in XRL hanging drop plates (MD3-24) and incubated at 4 °C.

*apo WRN(517-941):* Mother liquor was formed of 100 mM BIS-TRIS pH 5.8, 25% w/v PEG3350 and 200 mM LiSO$_4$. 200 nL drops were set at a 2:1 WRN$^{517-941}$ to mother liquor ratio; crystals grew and were harvested after 3 days, using 20% ethylene glycol for cryo-protection.

*WRN(517-941) + ATPγS + DNA:* WRN$^{517-941}$ was incubated with 0.3 mM ssDNA (sequence used: CGTACCCGATGTGTT), 5 mM ATPγS and 5 mM MgCl$_2$ on ice for 2–4 h. Crystals grew in 100 mM HEPES pH 7.2 and 15% w/v PEG 8 K within 24 h and were harvested after 2 days using 20% ethylene glycol for cryo-protection.

*WRN(517-941) + molecule 81 (VVD):* crystals were grown following incubation of WRN$^{517-941}$ with 2 mM of molecule 81 on ice for 2 h, in 100 mM BIS-TRIS 5.8 pH, 26% w/v PEG 3350 and 275 mM LiSO$_4$. Crystals appeared within 12 h and were harvested after 3 days using 20% ethylene glycol used for cryo-protection.

*WRN + HRO761 and WRN + GSK_WRN3:* crystals were grown following incubation of WRN$^{517-943}$ with 2 mM ligand in 2% Dimethyl Sulfoxide (DMSO), and incubated on ice for 2 h in 0.3 M trisodium citrate tetrahydrate, 0.1 M sodium citrate, pH 4.25 and 10% PEG 3350. Crystals emerging after 3 days. For GSK_WRN3, crystal growth was seeded by streak seeding with WRN + HRO761 co-crystals.

Data were collected at DLS on I04-1, I04 or I03, at ESRF on ID30A, or at MaxIV on BioMAX.

Single-crystal data were processed with autoPROC[40], using the STARANISO[41] scaling procedure, and molecular replacement performed with PHASER[42]. Ligand restraint generation was done with Grade2[43]. Refinement rounds with BUSTER[44] were interspersed with model building in COOT[45], until it was assessed that refinement had converged. Figures were generated using Pymol or Chimera. Structures have been deposited in the PDB as: WRN in complex with ATPgS and ssDNA (**9S19**); apo WRN helicase (**9S1A**), WRN helicase in complex with molecule 81 (**9S17**), WRN in complex with GSK_WRN3 (**9S1B**), WRN helicase in complex with HRO761 (**9S18**).

### SPR
SPR experiments were carried out using a Biacore 8K+ instrument (Cytiva) with WRN$^{517-941}$ protein. For measurements of binding to DNA, a 15-mer oligonucleotide (5′-Biotin-TEG-CGTACCCGATGTGTT-3′) was captured on a Neutravidin-coated sensor chip by injecting 0.05 μM oligonucleotide for 12 s at 5 μL/min onto the primed surface, to give a capture level of 50 RU. Experiments were carried out in a running buffer (10 mM HEPES pH 7.4, 150 mM NaCl, 0.005% P20, 5 mM MgCl$_2$, 1 mM TCEP) supplemented with 2% DMSO when necessary. Analysis was conducted at temperatures of 20 °C in the flow cell and 10 °C in the sample compartment. WRN protein was diluted to 10 μM in run buffer supplemented with NSB

reducer (1 mg/mL) in the absence or presence of cofactor or compound. Serial dilutions were prepared, varying concentration of protein, while maintaining concentration of other components. The system was equilibrated with 6× startup injections of run buffer for 40 s and 60 s dissociation time at 30 μL/min, then analyte solutions were injected for 60 s, 120 s dissociation at 30 μL/min.

For experiments with WRN protein captured on the sensor surface, WRN protein was expressed with an AviTag fused to the C-terminus via a GS-linker. The protein was prepared as above with the following modifications: co-expressed with BirA biotin ligase and expression media supplemented with biotin (40 μM), to generate 100% biotin-labelled protein by intact mass spectrometry. Using the same run buffer conditions as outlined above, WRN protein was captured on a Neutravidin chip by injecting a 30 μg/mL solution of protein for 600 s at 5 μL/min, resulting in a capture level of ~10,000 RU. The flow cell temperature was reduced to 10 °C to preserve the integrity of the protein, and the sample compartment was set at 10 °C. The Biacore A-B-A inject mode was used. A solution of single-stranded unmodified DNA oligonucleotide (15 μM), with the same sequence as above, in run buffer was used as a flanking solution (injection A), injecting both pre- and post-analyte for 60 s at 30 μL/min. For analyte solution (injection B), serial dilutions of ATPγS (top concentration of 50 μM) were prepared by diluting a run buffer containing 15 μM oligonucleotide DNA. Analyte solutions were injected for 60 s at a flow rate of 30 μL/min.

For experiments using compound solubilised in DMSO, the flow system was washed between injections with a solution of 50% DMSO and effects of DMSO were corrected using solvent correction. All data were double-referenced and processed using the Biacore Insight evaluation software (Cytiva).

### Biochemical assays
Assay buffer was 25 mM Tris-HCl pH 8.0, 2 mM MgCl$_2$, 5 mM NaCl, 1 mM DTT, and 0.01% Tween-20. Assay readouts were measured on the Pherastar FSX (BMGLabtech) plate reader. For IC$_{50}$ determination, serial dilutions of test compounds and/or nucleotides were dispensed into plates with a top concentration of 30 μM and a final DMSO concentration of 0.3% across all wells. Data were normalised to 0.3% DMSO (0% inhibition) and 10 μM HRO761 (100% inhibition) controls and plotted using either XLFit, VitroVivo software (Revvity) or GraphPad Prism (v10). All concentrations stated are final assay concentrations.

### Florescence polarisation (FP) assays
FP assays were performed using the bodipyFL-ATPγS (ThermoFisher) probe in 384-well black plates (Corning) with a final reaction volume of 20 μL.

For IC$_{50}$ determination, WRN (400 nM) and probe (6.25 nM) in assay buffer, were added to compound plates and incubated for 1 h at room temperature, protected from light, before measuring FP signal with the FP 485 520 520 optic module.

For $K_D$ measurements, a titration of WRN (10 μM top diluted 1:2) in the presence of probe (6.25 nM) was prepared in assay buffer with or without 2.5 μM ssDNA (CGTACCCGATGTGTT). The plate was protected from light, and FP was measured after 1 h incubation at room temperature.

### Helicase assay
WRN and BLM helicase unwinding assays were adapted from Sommers et al.[46] and performed in 384 well black plates (Greiner) with a final reaction volume of 25 μL.

Briefly, the forked-DNA substrate, FORKF, was generated by boiling equal amounts of oligoA-BHQ2 (TTTTTTTTTTTTTTTTTTTTTTTTTTTTTTTCGTACCCGATGTGTTCGTTC-BHQ2) and oligoB-TAMRA (TAMRA-GAACGAACACATCGGGTACGTTTTTTTTTTTTTTTTTTTTTTTTTTTTTT) in 10 mM Tris-HCl (pH 8.0) and 5 mM NaCl for 5 min at 95 °C and allowed to cool slowly to room temperature.

## Table 1 | Data collection and refinement statistics

| | WRN in complex with ATPgS and ssDNA (9S19) | APO WRN helicase (9S1A) | WRN helicase in complex with molecule 81 (9S17) | WRN in complex with GSK_WRN3 (9S1B) | WRN helicase in complex with HRO761 (9S18) |
|---|---|---|---|---|---|
| Space group | P21 | P21 | P21 | P212121 | P212121 |
| a, b, c (Å) | 62.42, 70.55, 123.47 | 70.76, 80.24, 83.09 | 56.091, 92.729, 96.298 | 68.850, 71.938, 121.555 | 67.177, 68.159, 119.009 |
| alpha, beta, gamma (°) | 90.000, 94.898, 90.000 | 90.00, 90.27, 90.00 | 90.000, 105.568, 90.000 | 90.0, 90.0, 90.0 | 90.0, 90.0, 90.0 |
| Resolution (Å)* | 123.019–2.300 (2.373–2.300) | 57.719–1.956 (2.005–1.956) | 92.765–1.910 (2.108–1.910) | 61.909–2.218 (2.745–2.218) | 59.505–1.995 (2.150–1.995) |
| Rpim overall* | 0.035 (0.377) | 0.031 (0.439) | 0.031 (0.439) | 0.058 (0.317) | 0.021 (0.490) |
| I/σ(I)* | 13.9 (2.1) | 15.9 (1.8) | 12.9 (1.7) | 11.3 (1.3) | 17.4 (1.5) |
| CC1/2* | 0.999 (0.412) | 0.999 (0.423) | 0.999 (0.674) | 0.998 (0.427) | 0.999 (0.636) |
| Completeness (%)* | 87.5 (49.6) | 90.9 (70.4) | 94.4 (69.3) | 82.7 (17.2) | 94.5 (63.3) |
| Multiplicity* | 6.2 (4.9) | 6.7 (6.1) | 7.0 (6.1) | 11.5 (5.5) | 13.6 (14.3) |
| PDB Resolution range | 123.02–2.300 | 57.72–1.956 | 92.77–1.910 | 61.91–2.218 | 24.13–1.995 |
| PDB Reflections | 41,776 | 60,721 | 47,870 | 16,012 | 29,180 |
| PDB Rwork/Rfree | 0.2282/0.2565 | 0.2068/0.2472 | 0.2190/0.2476 | 0.2462/0.2939 | 0.2104/0.2509 |
| Bond length RMSD (Å) | 0.007 | 0.008 | 0.009 | 0.008 | 0.008 |
| Bond angle RMSD (Å) | 0.91 | 0.95 | 0.97 | 0.95 | 0.93 |
| Average B factor (Protein) | 57.32 | 40.12 | 52.36 | 63.45 | 53.36 |
| Average B factor (Ligand) | | | 55.75 | 58.13 | 41.47 |
| Average B factor (Solvent) | 51.22 | 44.51 | 51.7 | 45.72 | 62.32 |
| Number of atoms (Protein) | 6702 | 6268 | 6384 | 3304 | 3366 |
| Number of atoms (Ligand) | 0 | 0 | 50 | 24 | 49 |
| Number of atoms (Solvent) | 294 | 420 | 342 | 70 | 291 |
| Ramachandran (%) Favored | 96.39 | 97.55 | 97.09 | 96.37 | 97.86 |
| Ramachandran (%) Allowed | 3.36 | 2.45 | 2.91 | 3.63 | 2.14 |
| Ramachandran (%) Outliers | 0.25 | 0 | 0 | 0 | 0 |

*Values in parentheses are for the highest-resolution shell.

Compounds were pre-incubated for either 30 min or 4 h, with either WRN or BLM (10 nM) in assay buffer containing 2.5 μg/mL calf thymus DNA. FORKF (100 nM) and ATP (2 mM) were then added, plates were sealed, briefly centrifuged, and incubated at 25 °C for 60 min. Following incubation, fluorescence (Ex 520/Em 595) was measured and data analysed as outlined above.

### Cell culture

HCT116 and HT29 cells were sourced from ATCC and N-terminally HiBit-tagged WRN HCT116 cells were generated at WuXi HDB from a single clone. Knock-in was confirmed by sequencing. All cells were grown in McCoy's 5A (Modified) media supplemented with 10% heat-inactivated FBS, and maintained in a humidified incubator at 37 °C with 5% $CO_2$.

### Generation of resistant cell lines

Resistant HCT116 cell lines were established over a period of 3 months by continuous exposure to increasing concentrations of HRO761 or VVD-13321 (Supplementary Fig. 5), starting from the $IC_{50}$ dose in media. 0.1% DMSO was maintained throughout. If viabilities were low, media would be drug-free until the cells recovered. For a control, HCT116 cell line chronically treated with 0.1% DMSO was also maintained.

### Cellular assays

Compounds were pre-dispensed into the inner wells of 384-well plates using a Labcyte Echo 555 or Tecan D300E liquid handler. Compounds were serially diluted, 1:3, with a maximum concentration of 30 μM. DMSO (0.3%) was normalised across the plates. DMSO only, staurosporine (10 μM) and HRO761 (10 μM) controls were included on every plate. Assay readouts were measured on the Pherastar FSX plate reader, and data were analysed using VitroVivo software.

### Cell viability assay

HCT116 (500 cells/well) and HT29 (700 cells/well) cells were dispensed into compound plates (clear-bottom white) using a Multidrop Combi Reagent dispenser (ThermoFisher). Cells were also dispensed into an additional plate at the same seeding density to use as a day 0 ($T_0$) readout. Cells were allowed to settle for 20 min at room temperature, and additional PBS was added to the outer wells to minimise evaporation, before transferring to a humidified incubator at 37 °C with 5% $CO_2$. Plates were harvested by dispensing 25 μL of CellTiter-Glo 2.0 (Promega) pre-warmed to room temperature, shaken for 1 min followed by a 30 min incubation at room temperature protected from light, before measuring luminescence. The day 0 plate was harvested 20 min to 1 h following addition of cells, treated cells were incubated for

5 days. Percent inhibition was calculated using $T_0$ as 100% inhibition and DMSO-only wells as 0% inhibition.

## RNA extraction and RNAseq
One million cells of 3 biological replicates were washed with PBS and snap-frozen in dry ice. Samples were collected from HCT116, 0.1% DMSO treated HCT116, HRO761 and VVD-133214 chronically treated HCT116 in the early (HRO761: 1 μM; VVD-133214: 600 nM) and late (HRO761: 13 μM; VVD-133214: 3 μM) stages of resistance development, and HRO761 and VVD-133214 acute treated HCT116. For acute treatment, the 0.1% DMSO HCT116 cell line was grown in 13 μM HRO761 or 3 μM VVD-133214 for one passage.

RNA was isolated and purified from snap-frozen cells using the RNeasy Mini Kit (Qiagen) according to the manufacturer's instructions. The purification included a DNase treatment using the RNase-free DNase Set. Purified RNA was submitted to the Genomics Facility at the Babraham Institute for library preparation and RNAseq. Overall RNA quality was assessed using the Agilent TapeStation (RIN scores > 9). cDNA libraries were constructed using Watchmaker DNA Library Prep Kits. Libraries were quantified using qPCR. Samples were sequenced using AVITI with High Output to generate 150 bp reads.

Raw fastq files from the sequencer were trimmed with Trim Galore v0.6.10[47] using default parameters to remove adapters and poor-quality base calls. Trimmed fastq files were then indexed and aligned to the human transcriptome GRCh38.p14 v47[48] using Salmon v1.10.0[49]. Transcript IDs were matched to gene IDs from the human genome GRCh38.p14 v47[48]. Data wrangling and manipulation was done in JuypterLab v4.0.7[50] using Python v3.11.12[51] and pandas v2.2.3[52]. Transcripts per million was used for normalisation. Differential gene expression analysis was performed using pyDESeq2 v0.5.1[53]. 0.1% DMSO treated HCT116 and treatment-naive HCT116 cell lines were grouped as a control and compared to HRO761 and VVD-133214 treated samples. Log2fold changes of WRN were plotted in Graphpad prism.

To identify mutations, the Babaham bioinformatics facility trimmed and aligned fastq files to the human GRCh38 genome assembly using hisat2 v2.2.1[54] guided by gene modes from Ensembl v87[55] and performed mapping using default parameters with soft clipping disabled. BAM files were sorted and indexed by pysam v0.23.0[56,57] in JuypterLab v4.0.7[50]. BAM and associated BAM.BAI files were then used in igv-notebook v3.1.4[58,59].

## Protein extraction and western blot
HCT116, 0.1% DMSO treated HCT116, HRO761 and VVD-133214 chronically treated HCT116 in the early (HRO761: 1 μM; VVD-133214: 600 nM) and late (HRO761: 13 μM; VVD-133214: 3 μM) stages of resistance establishment, and HRO761 and VVD-133214 acute treated HCT116 cells were seeded in 6-well plates (1 M cells/well). For acute treatment, the 0.1% DMSO HCT116 cell line was grown in 13 μM HRO761 or 3 μM VVD-133214.

Plates were incubated in a humidified incubator at 37 °C with 5% $CO_2$. After 3 days, cells were washed twice with ice-cold PBS before addition of 200 μL of ice-cold 1X cell lysis buffer (Cell Signalling Technology) containing 1X PhosSTOP™ (Roche), 1X cOmplete™ ULTRA Protease Inhibitor Cocktail (Roche) and BaseMuncher Endonuclease (Abcam). Cells were scraped and collected in an ice-cold Eppendorf tube. Samples were agitated for 30 min in a cold room, then centrifuged at $12,000 \times g$ for 20 min at 4 °C and supernatant was recovered. Protein content was quantified using the Pierce™ BCA Protein Assay (ThermoFisher Scientific).

For western blotting, samples were prepared by adding 1x NuPAGE™ LDS Sample Buffer (Invitrogen), supplemented with DTT (Invitrogen) and then boiled for 5 min at 95 °C. Protein (10 μg) was loaded onto 4–20% Mini-PROTEAN® TGX™ Precast Protein Gels (Bio-rad) and then separated using Sodium Dodecyl Sulfate Polyacrylamide Gel Electrophoresis (SDS-PAGE). Proteins were transferred to a Polyvinylidene fluoride (PVDF) membrane (Trans-Blot Turbo Mini 0.2 μm PVDF Transfer Packs, Bio-rad). Membranes were blocked with 5% w/v non-fat Milk (Melford Laboratories) in Tris-buffered Saline with Tween20 and incubated with 1:1000 WRN (8H3) Mouse monoclonal antibody. Membranes were washed and then incubated with 1:3000 Anti-mouse IgG HRP-linked polyclonal antibody (Cell signalling Technologies) and 1:1000 HRP-conjugated Alpha Tubulin Monoclonal antibody (ProteinTech). Protein bands were visualised by the addition of Clarity Western ECL Substrate (Bio-Rad) and captured using Fusion FX (Vilber).

## Data analysis, statistics and reproducibility
For biochemical and cellular assay data reported averages were calculated from the $-\log_{10}$-transformed $XC_{50}$ values and are presented as mean ± standard deviation (SD) from n independent replicates, as specified in Supplementary Table 1. Individual data points are shown in all graphs to illustrate data distribution and number of independent technical replicates are stated in the figure legend.

SPR experiments were repeated independently at least twice with similar results. Figures show representative sensorgrams, and the underlying numerical source data are provided in the Supplementary Data file.

Crystallographic structures were each determined from a single dataset. Data-collection and refinement statistics are provided in Table 1, and coordinates and structure factors have been deposited in the Protein Data Bank under the accession codes listed in the manuscript.

## Reporting summary
Further information on research design is available in the Nature Portfolio Reporting Summary linked to this article.

## Data availability
Crystallographic data have been deposited in the Protein Data Bank (PDB) under accession codes 9RUS, 9RUR, and 9RTI. Data-collection and refinement statistics are provided in Table 1. The numerical source data for all graphs and charts, including biochemical and cellular assays and surface plasmon resonance (SPR) experiments, are provided in the Supplementary Data File 1. All other data supporting the findings of this study are available from the corresponding author upon reasonable request. The RNA-seq data generated in this study has been deposited in the Gene Expression Omnibus (GEO) with accession number GSE314786. Raw sequencing reads (FASTQ files) and processed count tables are available through this repository, and differential expression results are provided in the Supplementary Data File 2. Uncropped and unedited blot/gel images are presented in Supporting Data Fig. 1 within the Supplementary Information File.

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

## Acknowledgements
We thank the In Vitro Biology team at WuXi AppTec, specifically Xiuhua Zhang, Yan Kong and Jingjing Chen, for performing and analysing routine biochemical and cellular screening assays to assess compound activity. We are also grateful to the structural biology team at Crelux GmbH (WuXi AppTec) for their insights into the characterisation of the 'twisted' WRN conformation. We also acknowledge the Genomics Facility at the Babraham Research Institute, in particular Megan Hamilton, for running the sequencing experiments and providing expertise in experimental design; and the Bioinformatics Group at the Babraham Research Institute, led by Simon Andrews, for support with processing and analysis of large-scale sequencing datasets.

## Author contributions
C.T.F. designed and performed structural experiments, acquired and analysed crystallographic data, interpreted results, and co-wrote the manuscript. A.A.M. designed and carried out biochemical and cellular experiments, analysed data, interpreted findings, and co-wrote the manuscript. C.B. developed and analysed resistance cell lines, performed RNA-seq and Western blot experiments, and co-wrote the manuscript. P.C. designed and performed SPR experiments, analysed and interpreted data, and co-wrote the manuscript. P.R. and A.S.D. analysed and interpreted structural data; P.R. also reviewed the manuscript. S.H. and C.D.O.C. designed, executed, and analysed protein production experiments; C.C.G.S ran and analysed molecular dynamic simulations; D.R. contributed to study conception and data interpretation; G.M.T.S. and S.E.S. provided project leadership and contributed to study conception and data interpretation; S.E.S. also reviewed the manuscript. A.J.K. provided project leadership, including conception, design, execution, and interpretation of data, and was responsible for the development of the manuscript, writing, and final review.

## Competing interests
All authors were employees of CHARM Therapeutics at the time of this work. CHARM previously pursued WRN inhibitor development, the program is no longer active, and this work was written up after its termination. Some authors are minor shareholders of CHARM Therapeutics, all other authors declare no competing interests.
