## [Transparent Peer Review file · Communications Biology]

Structural insights into WRN helicase reveal conformational states and opportunities for MSI-H cancer drug discovery

Corresponding Author: Dr Amanda Kennedy

Version 0:

Reviewer comments:

Reviewer #1

(Remarks to the Author)

The manuscript by Fletcher C. and co-authors describes the conformational dynamics of WRN helicase that drives DNA unwinding upon binding to DNA and nucleotides. In this paper the authors have demonstrated the conformational changes that favors or disfavors DNA and nucleotide binding to WRN helicase. Crystal structures of apo WRN or WRN bound to inhibitors provide further insight into the conformational requirement for DNA binding. The authors state that the flexibility of the helicase core modulates the relative orientation of the D1 and D2 domains that governs the DNA binding and subsequent unwinding reactions. The authors have shown that the inhibitor bound WRN can adopt a 'twisted' conformation that is incompetent in DNA binding. The authors have also tried to understand the resistance mechanism for known WRN inhibitors. Their overall hypothesis is fairly supported by good quality structural data and biochemical and biophysical evidences. However, I have a couple of concerns and few suggestions.

1. Page 5, line 121-122: From the domain displacement observed in the WRN crystal structures in presence of inhibitors, authors conclude that the D1 and D2 domains are highly flexible and can rotate freely around each other. Crystal structures in absence or presence of inhibitors can provide limited information about the conformational flexibility and typically represents the snapshot of predominant conformational state. I suggest to rephrase the statement and highlight on the specific conformations (with GSK_WRN3) observed in their case. Moreover, it would be informative to perform Molecular dynamics simulations with the WRN structure to gain insight into domain motions and conformational flexibility.
2. In the SPR experiments, authors show that in the presence or absence of non-hydrolysable ATP analogue, WRN binds to ssDNA with similar affinity while in presence of ADP there is severe loss of DNA binding affinity. This would mean that the ADP bound open conformation disfavors DNA binding while the ATP-gamma-S bound state represents a closed conformation similar to the apo state and favors DNA binding. The authors need to clearly explain the conformational states of WRN in the nucleotide catalytic cycle. SPR or fluorescence polarization experiments for ADP binding with WRN under saturating ssDNA concentration will be interesting and informative.
3. The DNA binding experiments were performed only in presence of ssDNA. It would be informative to repeat experiments with dsDNA duplex or branched DNA substrates that are physiologically more relevant for WRN helicase.
4. The authors report a very interesting finding regarding the ARL structural transition from unstructured loop to alpha helix upon DNA binding. This is in agreement with previous reports where ARL has been shown to be the major structural element that couples ATP hydrolysis to DNA binding in SF2 helicases. Recent report by Das et al., 2025 NAR as well as results from Windgassen et al., 2016 NAR have demonstrated similar allosteric regulation of nucleotide binding and hydrolysis upon DNA binding to ARL in RECQ1 and PriA helicases respectively. It would be relevant to discuss these papers in context of the findings reported in this manuscript.
5. Authors have identified polar interactions of the phosphate backbone with R774 and T821 on the D2 domain. It seems like these contacts are essential for DNA binding and ensuring proper interaction with the ARL and coupling DNA binding to ATP hydrolysis. Mutational analysis of these two residues followed by DNA stimulated ATPase experiments will provide direct evidence in support of their hypothesis.
6. The authors have performed DNA binding experiments after pre-incubation with HRO671 or GSK_WRN3 to infer that DNA binding is inhibited in presence of these inhibitors. Did the authors try to assess the DNA dependent ATPase activity in presence of the inhibitors?
7. Page 12: line 318: The authors propose that conformational locking of WRN due to inhibitor binding may be recognized as misfolded protein by the degradation machinery. This is unlikely and needs experimental validation. I would suggest to remove or modify this statement.

Reviewer #2

(Remarks to the Author)

Summary:

Fletcher et al. have investigated the structural biology aspects of the WRN helicase implicated in not only a premature aging disorder known as Werner syndrome but also a synthetic lethal target in microsatellite instability high (MSI-H) cancers. In this study, protein crystallization of a recombinant WRN helicase domain fragment was used to solve its structure both in apo form and bound to ssDNA. In addition, biochemical and biophysical characterization of the WRN fragment with previously published clinical WRN helicase inhibitors were performed to gain new insights into the effects of the compounds on WRN conformational state. Finally, a limited set of cell-based assays was performed to assess outcomes from chronic treatment. A single WRN mutation corresponding to an amino acid substitution (F730S) richly emerged in the cell population from chronic exposure to WRN inhibitor HRO761; this residue resides on the edge of the drug binding site in WRN. The authors also identified a WRN point mutation (T705A) that arose in the cell population with chronic exposure to VVD-133214. Based on the collective data presented, the authors propose a “WRN lifecycle” model for drug inhibitor mechanism involving protein conformational changes in WRN involving nucleotide and DNA.

Overall Critical Comments:

Overall, this study carries value in that it encompasses novelty (new WRN structures with or without ssDNA), as well as insights to nucleotide-induced changes to WRN conformation and the induced effect of clinical WRN helicase inhibitor drugs, and observations pertaining to drug resistance in cellulo via WRN mutation. The findings are significant and of considerable interest to not only specialists in the helicase community but also cancer biologists and aging researchers who consider Werner syndrome the model genetic disease of premature aging. With some significant improvements (see below, Specific Comments), the manuscript may be worthy of further consideration for publication.

Specific Comments:

Introduction: It is extremely odd and, in my mind, inappropriate to omit mention that bi-allelic mutations in WRN are linked to a hereditary premature aging disorder and provide the corresponding citation of that discovery (PMID: 8602509). This should appear at the outset of Introduction.

Introduction: Immediately adjacent to the first mention of “HRO761” and “VVD-133214”, the references should be provided.

Results—line 156 “Given that WRN is a non-specific helicase” This statement needs to be struck or replaced with a more accurate and meaningful description of WRN as a helicase enzyme. What is written is not only confusing but tremendously misleading.

Results: While the authors examined the effect of the WRN helicase inhibitors on WRN protein degradation in HCT116 cells, they conspicuously omitted an assessment of drug inhibitor induced WRN protein chromatin enrichment, a good possibility given previous findings from previous WRN helicase inhibitor studies. Given the interest in static drug-protein-DNA complexes as a source of cytotoxicity for WRN and other DNA-interacting factors (e.g., PARP), I found it surprising that this topic was not addressed in the study.

Results: A single WRN mutation corresponding to an amino acid substitution (F730S) richly emerged in the cell population from chronic exposure to WRN inhibitor HRO761; this residue resides on the edge of the drug binding site in WRN. The authors also identified a WRN point mutation (T705A) that arose in the cell population with chronic exposure to VVD-133214. The authors might have tested if the purified recombinant WRN harboring either the F730S or T705 missense mutations rendered the mutant proteins deficient in binding HRO761 and VVD-133214, respectively. If positive results would have been obtained, this would have greatly strengthened the hypothesis that either or both mutations impact WRN drug interaction. It would also have been of interest to determine if either drug-induced WRN missense mutation affects the protein’s biochemical activities (e.g., helicase).

Results: The authors suggest that the reduced potency of VVD-133214 chronic cellular treatment may be in part attributed to up-regulated WRN protein expression. This hypothesis could be directly tested by transfecting the cells with wild-type WRN and assessing if exogenous WRN expression suppressed the cytotoxic effects of VVD-133214.

Abstract, Discussion, perhaps elsewhere in text: “WRN life cycle” is odd terminology as presented in abstract, Figure 6 figure legend, and perhaps elsewhere to describe WRN conformational changes induced by nucleotide / DNA / inhibitor binding. Recommend changing this terminology because WRN itself is a protein, not a biological organism. Perhaps “structural cycle”, a term used in the Discussion is more fitting.

Reference Previously Published Papers Relevant to Current Study: In several places within the manuscript, statements are made about others’ findings but those papers are not referenced when they are mentioned. This is a problem. I provide a few examples below, but the omissions of relevant previously published work is a significant and recurring problem throughout the manuscript. Therefore, the manuscript must be carefully examined for such omissions and the relevant references added.

- For example, lines 172-174: “The aromatic side chain of F680 is intercalated between the ssDNA bases, forming π -stacking interacting, a structurally conserved motif observed in other RecQ helicases, such as BLM.” The BLM paper mentioned at the end of the sentence should be referenced.

- Another example is found lines 58-59 “While other RecQ helicase structures have been reported...” Those papers should be cited here.

- Another example, found on lines 72-73: "While previously solved structures have captured WRN in isolated nucleotide-bound states, the DNA-bound conformation has remained elusive." The nucleotide-bound state WRN structure paper(s) should be cited here. The reader should not have to dig somewhere else in the paper, or even worse, have to dig in the literature.

- Another example: lines 219-220 "HRO761 and VVD-133214 are potent WRN inhibitors that lead to increased dsDNA breaks cell death selectively in MSI-H cell lines." Cite the papers!

Discussion: A relevant paper from the Keck lab (PMID: 16340008) presented data suggesting that the conserved aromatic-rich motif III of *E. coli* RecQ is important for coupling ATP hydrolysis to DNA interaction and duplex DNA unwinding. Does the structural work in the current study suggest conformational changes of WRN induced by ssDNA or nucleotide binding, affected by inhibitor, operate by a mechanism involving similar behavior as aromatic loop in RecQ?

Methods: Under RNA extraction and RNAseq, is "IM cells" one million cells?

Reviewer #3

(Remarks to the Author)

Small-molecule inhibitors targeting the helicase core of Werner syndrome helicase (WRN) are promising therapeutic agents for treating microsatellite instability–high (MSI-H) cancers. Following several impactful studies that elucidated the mechanism of such inhibitors, Fletcher et al. now present the first WRN helicase core structure bound simultaneously to ssDNA and an ATP analog. Unlike the inhibitor-stabilized "twisted" inactive state, the ssDNA-bound structure reveals a previously underappreciated conformation of the helicase core. This not only provides new insights into the helicase's DNA translocation mechanism but also offers opportunities for future structure-based drug design targeting the "DNA-on" state of the enzyme. Together with a systematic analysis of available WRN core structures (including four additional apo or inhibitor-bound states reported here), detailed investigation of helicase–DNA and helicase–inhibitor interactions, and in-cell studies of drug-induced resistance, this work delivers valuable insights for future drug development targeting WRN.

The manuscript is already concise and comprehensive. I have several comments and suggestions that I hope will help further strengthen the study:

Major Comments:

1. Figure 1D suggests that ADP prevents WRN from binding to ssDNA. What could be the structural basis for this inhibition? Would a comparison between the ADP-bound structure (PDB: 6YHR) and the WRN–ATPyS–ssDNA complex (PDB: 9S19) help explain this phenomenon?
2. In Figure 2E and F, ssDNA association does not appear to substantially affect WRN's affinity for ATPyS (6 vs 5 μ M). However, in Supplementary Figure 3, the affinity for ATP (Km values) seems markedly affected by the presence of thymus DNA (16.52 vs 42.77 μ M). Could the authors clarify this inconsistency in the text?
3. Related to the above, do the authors expect WRN514–914 (D1D2 only) to bind and unwind thymus DNA? Could they provide results using a forked-DNA substrate instead of thymus DNA for measuring ATP turnover? If this is not feasible, I would recommend at least explaining the rationale for using thymus DNA in the DNA-stimulated ATP hydrolysis assays.
4. The third Results section is titled "helicase activity," and the Materials and Methods include a section on "helicase assay." However, I was unable to find results directly measuring helicase activity of the proteins studied. While DNA-stimulated ATP hydrolysis (measured by ADP-Glo) may reflect helicase activity of WRN514–914, it would strengthen the study if the authors could include helicase activity assays in the Supplementary Data—ideally comparing results in the absence and presence of the inhibitors used here.
5. In lines 194–197, the text states: "WRN_GSK3 had a differentiated mechanism of inhibition." However, the Figure 4B legend states: "inhibitor concentration response curves show differentiated mechanisms of inhibition for VVD-133214 compared to HRO761 and GSK_WRN3." This is confusing. Based on Figure 4B, it appears that HRO761 and GSK-WRN3 displace ATPyS probe from WRN because their binding sites overlap with the nucleotide-binding site. In contrast, VVD-133214 binds to an alternative site, allowing it to associate with WRN even in the presence of ATPyS. Thus, VVD-133214 acts cooperatively with nucleotide, unlike typical WRN-targeting inhibitors. I recommend clarifying this discrepancy between the text and figure legend.
6. Is the closed conformation described in line 213 (trapped by VVD-133214/molecule 81) essentially the same as the closed conformation mentioned for the apo structure (line 94, described as a previously unrevealed state)? If not, please provide a more precise definition so readers can appreciate the differences.
7. The difference maps for the bound inhibitors shown in the PDB validation reports suggest some discrepancies between the modeled ligands and the experimental data. In addition, there appear to be a considerable number of side-chain outliers across the protein models. I recommend that the authors address these issues where feasible, for example by refining the ligand fits and correcting side-chain conformations.

Minor points:

1. Line 97: redefine the range of the flexible hinge as residues 728–734 (the loop region lacking β -structure), rather than 725–735.
2. Line 146: revise "Supplementary Figure 1A" to "Supplementary Figure 1."
3. Line 178: revise "1.5A" to "1.5 Å."

4. Line 197: revise “WRN_GSK3” to “GSK_WRN3.”

5. Lines 256–257: revise “in either the presence of absence of ssDNA” to “in either the presence or absence of ssDNA.”

6. Line 467: add a reference citation for Sommers et al.

Version 1:

Reviewer comments:

Reviewer #1

(Remarks to the Author)

The authors have addressed most of the queries satisfactorily. I am still not convinced with the explanation provided for reduced DNA binding in presence of ADP in contrast to ATP or ATPgammaS states. The authors state that there is an outward swing in the helices encompassing the residues 870-904 contributing to reduced DNA binding in ADP bound form of WRN. An overlay of ATP bound WRN (7GQT) and ADP bound WRN (6YHR) exhibits a perfect alignment of the domains with very low rmsd. The Kd for ssDNA binding in presence of ADP is more than 20 folds higher than ATP bound form and 400 folds higher than ATP-gammaS bound state. The structural basis for so much variation in ssDNA binding is not very clear. Authors need to address this.

I have no further queries and the manuscript has improved satisfactorily.

Reviewer #2

(Remarks to the Author)

The revised manuscript is much improved compared to the originally submitted manuscript. Although the authors did not perform all the requested experiments for the revision, they have clarified a number of confusing points and addressed a number of critical concerns.

Reviewer #3

(Remarks to the Author)

The authors have addressed all of my concerns. One minor suggestion: include definitions for pIC50, pEC50, pgIC50, and pDC50 in the footer of the new SI Table 1.

Fletcher et al. WRN manuscript

Reviewers' comments:

Reviewer #1 (Remarks to the Author):

The manuscript by Fletcher C. and co-authors describes the conformational dynamics of WRN helicase that drives DNA unwinding upon binding to DNA and nucleotides. In this paper the authors have demonstrated the conformational changes that favors or disfavors DNA and nucleotide binding to WRN helicase. Crystal structures of apo WRN or WRN bound to inhibitors provide further insight into the conformational requirement for DNA binding. The authors state that the flexibility of the helicase core modulates the relative orientation of the D1 and D2 domains that governs the DNA binding and subsequent unwinding reactions. The authors have shown that the inhibitor bound WRN can adopt a 'twisted' conformation that is incompetent in DNA binding. The authors have also tried to understand the resistance mechanism for known WRN inhibitors. **Their overall hypothesis is fairly supported by good quality structural data and biochemical and biophysical evidences.** However, I have a couple of concerns and few suggestions.

1. Page 5, line 121-122: From the domain displacement observed in the WRN crystal structures in presence of inhibitors, authors conclude that the D1 and D2 domains are highly flexible and can rotate freely around each other. Crystal structures in absence or presence of inhibitors can provide limited information about the conformational flexibility and typically represents the snapshot of predominant conformational state. I suggest to rephrase the statement and highlight on the specific conformations (with GSK_WRN3) observed in their case. Moreover, it would be informative to perform Molecular dynamics simulations with the WRN structure to gain insight into domain motions and conformational flexibility.

We thank the reviewer for this constructive comment. We agree that crystal structures represent discrete conformational snapshots, and that the flexibility inferred from these data should be interpreted within that context. The text has been revised (line 125-127) to clarify that we refer to conformational flexibility *between* observed structural states (e.g., inactive versus active conformations), rather than continuous motion within a single state. While inhibitor binding can lower the energetic barrier to certain conformations, allowing them to be captured crystallographically, these conformations must nonetheless be accessible to the protein. The observed $\sim 180^\circ$ rotation of the D2 domain relative to D1 supports hinge-mediated flexibility between these states. Molecular dynamics simulations were not performed, as capturing such large domain rearrangements would require extensive computational resources beyond the scope of this study.

2. In the SPR experiments, authors show that in the presence or absence of non-hydrolysable ATP analogue, WRN binds to ssDNA with similar affinity while in presence of ADP there is severe loss of DNA binding affinity. This would mean that the ADP bound open conformation disfavors DNA binding while the ATP-gamma-S bound state represents a closed conformation

similar to the apo state and favors DNA binding. The authors need to clearly explain the conformational states of WRN in the nucleotide catalytic cycle. SPR or fluorescence polarization experiments for ADP binding with WRN under saturating ssDNA concentration will be interesting and informative.

We thank the reviewer for this comment. We have clarified the relationship between nucleotide state, domain orientation, and DNA affinity in the Discussion.

Added text at line 275:

'We refer to the 'closed' conformation as the state where the angle between D1 and D2 is smallest; and 'open' conformation where the angle is increased. The ability for WRN to bind DNA has more complexity than angle between D1 and D2 alone; and is connected to the specific movements in the nucleotide binding site and D2 domain for which we now have structural insight and hence will refer additionally to 'on-DNA' and 'off-DNA' conformations independently.'

Additionally, we have clarified the difference between the Apo/ATPgS/ADP structures that offer an explanation for the biophysical data (see also Reviewer 3 comment 1)

Added text line 304:

'Direct overlays of apo, ATPgS+ssDNA-bound and ADP-bound WRN (PDB 6YHR⁷) suggest that when ADP is bound (in comparison to no nucleotide or ATPgS) there is an outward swing of the helices formed by residues 870-904. This movement may limit engagement of the ratchet and contribute to the reduced affinity for DNA observed; although further work would be required to explore this.'

FP experiments were carried out to investigate binding of ssDNA to WRN in the presence of the ATPgS probe. Surprisingly, we found that saturating conditions of ssDNA fully outcompeted the ATPgS probe. We believe that this is due to the overlap of the probe molecule attached to the ATPgS and DNA binding being incompatible in this system. As such it was not possible to carry out the FP experiment as suggested by the reviewer - without dye binding in the presence of ssDNA we cannot investigate ADP binding. The SPR experiment as described, we agree would also be informative, however also presented challenges to delivering results. With WRN attached to the SPR chips, when DNA was flowed over we saw up to 40% binding, therefore measurement of ADP binding in this system would have the compounding factor of ADP binding to WRN alone. Therefore we do not believe that it is possible, or additive, to include the requested experiments in this manuscript.

3. The DNA binding experiments were performed only in presence of ssDNA. It would be informative to repeat experiments with dsDNA duplex or branched DNA substrates that are physiologically more relevant for WRN helicase.

For the purposes of this work, a ssDNA substrate was used as a surrogate to the more complex DNA structures that have unpaired regions, such as flap-strands or replication fork intermediates, to which WRN is known to bind. To support the analysis of the structural lifecycle of WRN in relation to the crystal structures, the authors felt it was most relevant to use ssDNA across the different studies. Additional experiments such as in the helicase assay used a forked DNA structure and this data is present in Supplementary Table 1.

4. The authors report a very interesting finding regarding the ARL structural transition from unstructured loop to alpha helix upon DNA binding. This is in agreement with previous reports where ARL has been shown to be the major structural element that couples ATP hydrolysis to DNA binding in SF2 helicases. Recent report by Das et al., 2025 NAR as well as results from Windgassen et al., 2016 NAR have demonstrated similar allosteric regulation of nucleotide binding and hydrolysis upon DNA binding to ARL in RECQ1 and PriA helicases respectively. It would be relevant to discuss these papers in context of the findings reported in this manuscript.

Discussion of mutational analysis of ARL in both SF2 helicases has been added to manuscript - authors thank reviewer for providing these useful and interesting references.

Text added to Line 292:

'This mechanism is observed in BLM helicase¹⁸ and is consistent with mutational studies of residues within the ARL in other RecQ^{10,14} and related SF2 helicases¹⁵ that suggest there is allosteric communication between the ARL and nucleotide binding site.'

5. Authors have identified polar interactions of the phosphate backbone with R774 and T821 on the D2 domain. It seems like these contacts are essential for DNA binding and ensuring proper interaction with the ARL and coupling DNA binding to ATP hydrolysis. Mutational analysis of these two residues followed by DNA stimulated ATPase experiments will provide direct evidence in support of their hypothesis.

We thank the reviewer for this suggestion and agree that additional experiments would provide direct evidence of the hypothesis presented and are an opportunity for future studies.

Unfortunately, it is not possible at this stage to perform additional experiments and so we hope that by highlighting the conserved nature of these residues across RecQ family members, will give the reviewer more confidence and support to the key role that they have in DNA binding.

Sequence alignment of RecQ helicases from multiple species shows a high degree of conservation for both residues (highlighted in blue box), supporting their functional relevance. Manuscript has been updated (line 165-176) and a panel showing this alignment added to Supplementary Figure 2B.

6. The authors have performed DNA binding experiments after pre-incubation with HRO671 or GSK_WRN3 to infer that DNA binding is inhibited in presence of these inhibitors. Did the authors try to assess the DNA dependent ATPase activity in presence of the inhibitors?

We thank the reviewer for this comment. We have assessed the effects of HRO671 and GSK_WRN3 on helicase activity of WRN. The compounds acted as potent inhibitors of WRN enzymatic activity, consistent with their strong inhibition of DNA binding. The helicase data is now provided in Supplementary Table 1.

7. Page 12: line 318: The authors propose that conformational locking of WRN due to inhibitor binding may be recognized as misfolded protein by the degradation machinery. This is unlikely and needs experimental validation. I would suggest to remove or modify this statement.

We thank the reviewer for this comment and agree that, while similar phenomena have been reported for other proteins in which inhibitor binding leads to destabilization and proteasomal degradation (for example, in the context of Hsp90 client proteins and small-molecule-induced protein degradation mechanisms), we do not have experimental evidence supporting such a process for WRN. As suggested, we have removed this speculative statement from the Discussion. As noted elsewhere, the degradation of WRN following inhibition in MSI-H cells is explained by activation of ATR/ATM-dependent ubiquitination pathways.

Reviewer #2 (Remarks to the Author):

Summary:

Fletcher et al. have investigated the structural biology aspects of the WRN helicase implicated in not only a premature aging disorder known as Werner syndrome but also a synthetic lethal target in microsatellite instability high (MSI-H) cancers. In this study, protein crystallization of a recombinant WRN helicase domain fragment was used to solve its structure both in apo form and bound to ssDNA. In addition, biochemical and biophysical characterization of the WRN fragment with previously published clinical WRN helicase inhibitors were performed to gain new insights into the effects of the compounds on WRN conformational state. Finally, a limited setoff cell-based assays was performed to assess outcomes from chronic treatment. A single WRN mutation corresponding to an amino acid substitution (F730S) richly emerged in the cell population from chronic exposure to WRN inhibitor HRO761; this residue resides on the edge of the drug binding site in WRN. The authors also identified a WRN point mutation (T705A) that arose in the cell population with chronic exposure to VVD-133214. Based on the collective data presented, the authors propose a “WRN lifecycle” model for drug inhibitor mechanism involving protein conformational changes in WRN involving nucleotide and DNA.

Overall Critical Comments:

Overall, **this study carries value in that it encompasses novelty (new WRN structures with or without ssDNA), as well as insights to nucleotide-induced changes to WRN conformation and the induced effect of clinical WRN helicase inhibitor drugs, and observations pertaining to drug resistance in cellulo via WRN mutation. The findings are significant and of considerable interest to not only specialists in the helicase community but also cancer biologists and aging researchers who consider Werner syndrome the model genetic disease of premature aging.** With some significant improvements (see below, Specific Comments), the manuscript may be worthy of further consideration for publication.

Specific Comments:

Introduction: It is extremely odd and, in my mind, inappropriate to omit mention that bi-allelic mutations in WRN are linked to a hereditary premature aging disorder and provide the corresponding citation of that discovery (PMID: 8602509). This should appear at the outset of Introduction.

We thank the reviewer for this comment. We agree that the link between biallelic WRN mutations and the premature aging disorder Werner syndrome is an important aspect of WRN biology, though the focus of the present study is on WRN as an oncology target rather than its role in aging. We have nevertheless added a brief mention of this genetic association, together with the recommended citation (PMID: 8602509), at the beginning of the Introduction for completeness.

Text added line 36:

Biallelic mutations in the human WRN gene cause Werner Syndrome, a rare hereditary premature aging disorder,¹ underscoring the essential role of WRN in genome maintenance.

Introduction: Immediately adjacent to the first mention of “HRO761” and “VVD-133214”, the references should be provided.

References added at first mention within the text.

Results—line 156 “Given that WRN is a non-specific helicase” This statement needs to be struck or replaced with a more accurate and meaningful description of WRN as a helicase enzyme. What is written is not only confusing but tremendously misleading.

Thank you for the reviewers request for clarification, this sentence has been rephrased (line 159).

‘Given that WRN shows structural specificity as opposed to rigid sequence specificity²², it is likely that there is variability in the bound portion of the ssDNA within the crystal, preventing absolute assignment of the ssDNA sequence, whereas the phospho-deoxyribose backbone was unambiguously modelled.’

Results: While the authors examined the effect of the WRN helicase inhibitors on WRN protein degradation in HCT116 cells, they conspicuously omitted an assessment of drug inhibitor induced WRN protein chromatin enrichment, a good possibility given previous findings from previous WRN helicase inhibitor studies. Given the interest in static drug-protein-DNA complexes as a source of cytotoxicity for WRN and other DNA-interacting factors (e.g., PARP), I found it surprising that this topic was not addressed in the study.

We thank the reviewer for this insightful comment. The current study focuses on clinically advanced WRN inhibitors, which, as demonstrated, do not bind to the WRN helicase core in the presence of DNA and therefore are unlikely to promote chromatin trapping of a productive WRN enzyme. For this reason, we did not assess WRN chromatin enrichment experimentally. However, we agree that the concept of inhibitor-induced WRN trapping on DNA is mechanistically important and could represent an effective strategy to overcome clinical resistance. We have expanded the Discussion to highlight that next-generation WRN inhibitors designed to engage the helicase core in its ‘on-DNA’ conformation could exploit a trapping mechanism analogous to that observed for PARP inhibitors and other DNA-processing enzymes (line 372), which could have clinically meaningful benefit.

Results: A single WRN mutation corresponding to an amino acid substitution (F730S) richly emerged in the cell population from chronic exposure to WRN inhibitor HRO761; this residue resides on the edge of the drug binding site in WRN. The authors also identified a WRN point

mutation (T705A) that arose in the cell population with chronic exposure to VVD-133214. The authors **might** have tested if the purified recombinant WRN harboring either the F730S or T705 missense mutations rendered the mutant proteins deficient in binding Hr0761 and VVD-133214, respectively. If positive results would have been obtained, this would have greatly strengthened the hypothesis that either or both mutations impact WRN drug interaction. It would also have been of interest to determine if either drug-induced WRN missense mutation affects the protein's biochemical activities (e.g., helicase).

We thank the reviewer for this thoughtful suggestion and fully agree that biochemical characterization of the F730S and T705A WRN variants would further strengthen the mechanistic link between these mutations and inhibitor resistance. Due to strategic changes within the organization, it was not possible to perform these follow-up experiments. As clarified in the Discussion (line 351-352), such studies will be important for future work to validate the precise impact of these mutations on compound binding and WRN helicase activity, but they are beyond the scope of the current manuscript.

Results: The authors suggest that the reduced potency of VVD-133214 chronic cellular treatment may be in part attributed to up-regulated WRN protein expression. This hypothesis could be directly tested by transfecting the cells with wild-type WRN and assessing if exogenous WRN expression suppressed the cytotoxic effects of VVD-133214.

We thank the reviewer for this suggestion. As described in the manuscript, WRN expression was indeed upregulated (>2-fold) in the late stages of resistance, which may contribute to the reduced potency of VVD-133214 (Figure 5F and Discussion). We are in agreement that functional validation of this mechanism through exogenous WRN expression would support this finding, however it would require a separate study design and was beyond the scope of this work. There is precedence in the literature that upregulation of protein targets can lead to a reduction in compound potency as stated in line 346.

Abstract, Discussion, perhaps elsewhere in text: "WRN life cycle" is odd terminology as presented in abstract, Figure 6 figure legend, and perhaps elsewhere to describe WRN conformational changes induced by nucleotide / DNA / inhibitor binding. Recommend changing this terminology because WRN itself is a protein, not a biological organism. Perhaps "structural cycle", a term used in the Discussion is more fitting.

We thank the reviewer for this suggestion and have implemented the change to structural cycle throughout the manuscript.

Reference Previously Published Papers Relevant to Current Study: In several places within the manuscript, statements are made about others' findings but those papers are not referenced when they are mentioned. This is a problem. I provide a few examples below, but the omissions of relevant previously published work is a significant and recurring problem throughout the

manuscript. Therefore, the manuscript must be carefully examined for such omissions and the relevant references added.

We thank the reviewer for this careful observation. We have thoroughly reviewed the manuscript to ensure that all statements referring to previously published work are appropriately cited. In addition to correcting specific examples highlighted by the reviewer, we have added or repositioned several references throughout the Introduction, Results, and Discussion to ensure comprehensive attribution of prior studies.

- For example, lines 172-174: “The aromatic side chain of F680 is intercalated between the ssDNA bases, forming π -stacking interacting, a structurally conserved motif observed in other RecQ helicases, such as BLM.” The BLM paper mentioned at the end of the sentence should be referenced.

This reference has been added

- Another example is found lines 58-59 “While other RecQ helicase structures have been reported...” Those papers should be cited here.

This reference has been added

- Another example, found on lines 72-73: “While previously solved structures have captured WRN in isolated nucleotide-bound states, the DNA-bound conformation has remained elusive.” The nucleotide-bound state WRN structure paper(s) should be cited here. The reader should not have to dig somewhere else in the paper, or even worse, have to dig in the literature.

This reference has been added

- Another example: lines 219-220 “HRO761 and VVD-133214 are potent WRN inhibitors that lead to increased dsDNA breaks cell death selectively in MSI-H cell lines.” Cite the papers!

We apologise for the lack of clarity here. In this case, the statement is part of the results section and in-house data is presented in Supplementary table 1, which has now been sign posted more clearly, and so it is not necessary to reference.

Discussion: A relevant paper from the Keck lab (PMID: 16340008) presented data suggesting that the conserved aromatic-rich motif III of E. coli RecQ is important for coupling ATP hydrolysis to DNA interaction and duplex DNA unwinding. Does the structural work in the current study suggest conformational changes of WRN induced by ssDNA or nucleotide binding, affected by inhibitor, operate by a mechanism involving similar behavior as aromatic loop in RecQ?

We thank the reviewer for this thoughtful question and have added a reference to the Keck lab study (PMID: 16340008) in the Discussion.

Text added on line 292:

This mechanism is observed in BLM helicase¹⁸ and is consistent with mutational studies of residues within the ARL in other RecQ^{10,14} and related SF2 helicases¹⁵ that suggest there is allosteric communication between the ARL and nucleotide binding site.

The conformational rearrangements observed in our WRN structures are consistent with large-scale domain movements modulated by nucleotide and ssDNA binding, but our data suggest a distinct mode of coupling compared with that proposed for *E. coli* RecQ. In our models, inhibitors such as VVD-133214 and HRO761 stabilize a ‘closed’ conformation of WRN, reducing the D1–D2 interdomain angle and thereby disrupting the trajectory of ssDNA across D2 and the conserved residues R774 and T821. This conformational locking explains the reduced ssDNA binding observed by SPR. While the aromatic-rich loop (ARL) of WRN likely remains capable of adopting structured conformations upon DNA engagement, inhibitor binding interferes with the interdomain geometry required for its proper positioning, consistent with impaired helicase activity.

Methods: Under RNA extraction and RNAseq, is “IM cells” one million cells?

This has been clarified in the text.

Reviewer #3 (Remarks to the Author):

Small-molecule inhibitors targeting the helicase core of Werner syndrome helicase (WRN) are promising therapeutic agents for treating microsatellite instability–high (MSI-H) cancers. Following several impactful studies that elucidated the mechanism of such inhibitors, Fletcher et al. now present **the first WRN helicase core structure bound simultaneously to ssDNA and an ATP analog. Unlike the inhibitor-stabilized “twisted” inactive state, the ssDNA-bound structure reveals a previously underappreciated conformation of the helicase core. This not only provides new insights into the helicase’s DNA translocation mechanism but also offers opportunities for future structure-based drug design targeting the “DNA-on” state of the enzyme.** Together with a systematic analysis of available WRN core structures (including four additional apo or inhibitor-bound states reported here), detailed investigation of helicase–DNA and helicase–inhibitor interactions, and in-cell studies of drug-induced resistance, **this work delivers valuable insights for future drug development targeting WRN.**

The manuscript is already concise and comprehensive. I have several comments and suggestions that I hope will help further strengthen the study:

Major Comments:

1. Figure 1D suggests that ADP prevents WRN from binding to ssDNA. What could be the structural basis for this inhibition? Would a comparison between the ADP-bound structure (PDB: 6YHR) and the WRN–ATPγS–ssDNA complex (PDB: 9S19) help explain this phenomenon?

This is challenging to structurally rationalise based on the available structural data, however comparison with existing structures for other RecQ helicases show a pattern of behaviour.

In both the apo WRN and WRN + ATPγS + ssDNA structures, the helices formed by residues 870-904 are stabilised at least in part by a hydrogen bonding network involving the highly conserved E846 and non-conserved, polar residues on the two helices (E844, K892, and Y891). Comparing to the ADP bound structure (PDB 6YHR) as the reviewer suggested, the helices are pointed in a more 'outward' direction and no Hydrogen bonds are formed between E846 and helical residues. This is consistent with the structural data on an alternative RecQ bound to DNA in the absence of nucleotide which is available. In the RecQ catalytic core from *C. sakazakii* bound to DNA crystal structure (PDB 4TMU), it adopts an 'open' pose as observed in the WRN + ATPγS + ssDNA, with inward movement of helices formed from residues 342-376 (equivalent to 870-904 in WRN), stabilised by Hbond network between E318 (conserved with E846) and the non-conserved, polar helical residue K367. And so the structural hypothesis as presented in the paper as to why apo shows similar binding in SPR as ATPγS-bound WRN, is because this interaction is still intact.

The SPR indicates that WRN + ADP binds ssDNA with significantly reduced affinity in comparison to either the apo or the ATPγS-bound forms. Again, whilst no structure of WRN + ADP + ssDNA exists to probe this directly, the structure of BLM + ADP + ssDNA (PDB 4CGZ), shows an 'outward' shift of the equivalent helices and no engagement of the ssDNA with the ARL, with the bases that would in theory interact with the ARL being disordered and unmodelled in the crystal structure. In contrast, the structure of BLM + ADP + allosteric inhibitor + ssDNA (PDB 7AUD) does show both engagement of the ARL with the ssDNA, and an inward shift of the equivalent helices, with an H-bond between the conserved Glutamic Acid (E971 in BLM, E846 in WRN) and non-conserved, polar, helical residues, further stabilised by direct interactions between the allosteric ligand and helical residues.

While the direct cause of inward / outward position of these identified helices is unclear based on these structures, the pattern of motions with engagement of the ssDNA to the ARL is consistent with the reported biophysical data and we hope that this has been clarified clearly in the manuscript.

Text added to manuscript (line 304):

Direct overlays of apo, ATPγS+ssDNA-bound and ADP-bound WRN (PDB 6YHR⁷) suggest that when ADP is bound (in comparison to no nucleotide or ATPγS) there is an outward swing of the helices formed by residues 870-904. This movement may limit engagement of the ratchet and contribute to the reduced affinity for DNA observed; although further work would be required to explore this.

2. In Figure 2E and F, ssDNA association does not appear to substantially affect WRN's affinity for ATP γ S (6 vs 5 μ M). However, in Supplementary Figure 3, the affinity for ATP (K_m values) seems markedly affected by the presence of thymus DNA (16.52 vs 42.77 μ M). Could the authors clarify this inconsistency in the text?

We thank the reviewer for this comment. The data in Figure 2E and F describe equilibrium binding of ATP γ S to WRN measured by SPR, whereas Supplementary Figure 3 presents steady-state kinetic parameters derived from WRN's ATPase activity. While the figure reports K_m and V_{max} , the text refers to the corresponding calculated k_{cat} values. These parameters reflect catalytic turnover rather than simple nucleotide binding affinity. The modest difference in K_m values (approximately threefold) falls within the expected experimental variation for biochemical assays and does not indicate a mechanistic inconsistency. Importantly, the increased k_{cat} observed in the presence of DNA supports enhanced ATP turnover during active translocation, consistent with our interpretation of DNA-stimulated WRN activity.

3. Related to the above, do the authors expect WRN514–914 (D1D2 only) to bind and unwind thymus DNA? Could they provide results using a forked-DNA substrate instead of thymus DNA for measuring ATP turnover? If this is not feasible, I would recommend at least explaining the rationale for using thymus DNA in the DNA-stimulated ATP hydrolysis assays.

Thank you for the reviewers observation. We have clarified in the supplementary methods why calf thymus DNA was used in the ATPase study and additionally added in the helicase inhibitor data in Supplementary Table 1, indicating that this protein construct can also unwind forked-DNA in a similar manner.

Text added to SI method:

Calf thymus DNA was used in these experiments to prevent interference of the luminescence readout with our dye-tagged forked DNA construct. As evident in the helicase assay data, this construct can also unwind forked-DNA and so it would be anticipated to show similar results.

4. The third Results section is titled "helicase activity," and the Materials and Methods include a section on "helicase assay." However, I was unable to find results directly measuring helicase activity of the proteins studied. While DNA-stimulated ATP hydrolysis (measured by ADP-Glo) may reflect helicase activity of WRN514–914, it would strengthen the study if the authors could include helicase activity assays in the Supplementary Data—ideally comparing results in the absence and presence of the inhibitors used here.

We thank the reviewer for this helpful suggestion. Direct measurements of WRN helicase activity were performed and are now included in the Supplementary Information (Supplementary Table 1). These data confirm that the tested compounds inhibit WRN helicase activity, consistent with the competition and cellular data described in the main text. The reason this is not exemplified further in the text is because the VVD-133214 molecule appears to be less potent biochemically than it is in cells. This is true under the conditions of the assays tested. Our helicase assay was optimised for the competitive mechanism demonstrated for HR0761 and

GSK_WRN3; rather than the cooperative mechanism exemplified by VVD-133214. As such the biochemical helicase readout is not accurate for this molecule, and does not align with the published data - the authors did not have the resource to re-optimize the helicase assay for alternative mechanism due to the strategy of the internal project, and feel that including this explanation in the main text does not enhance or add to the scope of the manuscript. It has been noted as a footer to the data in the Supplementary Table for clarity to the reader and we hope that this is satisfactory to the reviewer.

Text added to Figure legend:

* The apparent discrepancy between biochemical and cellular activity of VVD-133214 is due to the mechanism of the compound and the set-up of the biochemical assay, optimized for a competitive rather than cooperative mechanism of action, and as such the results reported for biochemical assays are artifactually lower for VVD-133214.

5. In lines 194–197, the text states: “WRN_GSK3 had a differentiated mechanism of inhibition.” However, the Figure 4B legend states: “inhibitor concentration response curves show differentiated mechanisms of inhibition for VVD-133214 compared to HRO761 and GSK_WRN3.” This is confusing. Based on Figure 4B, it appears that HRO761 and GSK-WRN3 displace ATPγS probe from WRN because their binding sites overlap with the nucleotide-binding site. In contrast, VVD-133214 binds to an alternative site, allowing it to associate with WRN even in the presence of ATPγS. Thus, VVD-133214 acts cooperatively with nucleotide, unlike typical WRN-targeting inhibitors. I recommend clarifying this discrepancy between the text and figure legend.

We thank the reviewer for identifying this discrepancy and for the clear summary of the binding behavior. We agree with the error and it is VVD-133214, rather than GSK_WRN3, that displays the differentiated mechanism of inhibition. The text in the Results section has been corrected to reflect this and now reads (Line 198):

Competition assays with the ATPγS FP probe demonstrated that despite VVD-133214 and GSK_WRN3 both targeting C727, VVD-133214 binds cooperatively in the presence of ATPγS (Figure 4B), in agreement with the literature¹³, while HR0761 and GSK_WRN3 exhibit overlapping binding with the nucleotide binding site.

This correction resolves the inconsistency between the text and Figure 4B.

6. Is the closed conformation described in line 213 (trapped by VVD-133214/molecule 81) essentially the same as the closed conformation mentioned for the apo structure (line 94, described as a previously unrevealed state)? If not, please provide a more precise definition so readers can appreciate the differences.

Yes, these are the same conformations; an additional line has been added to the manuscript in order to clarify the overlap between the APO and ligand bound conformations.

Text added to line 208:

The ligand bound structures show stabilisation of the Walker A motif across the nucleotide binding site, and a water-mediated hydrogen bonding network between D668, K577, T573 and the backbone carbonyl of M571, as well as the π - π stack between Y575 and Y849, a similar 'closed' conformation as apo WRN.

7. The difference maps for the bound inhibitors shown in the PDB validation reports suggest some discrepancies between the modeled ligands and the experimental data. In addition, there appear to be a considerable number of side-chain outliers across the protein models. I recommend that the authors address these issues where feasible, for example by refining the ligand fits and correcting side-chain conformations.

Minimal difference density peaks are observed at $F\sigma - Fc = 3\sigma$ for some modelled ligands, however the clear electron density for the $2F\sigma - Fc$ maps, as well as the low and stable B-factors post-refinement, support the ligands modelled as is.

WRN helicase is a highly flexible protein with multiple unstructured loops, and as such suffers from regions of broader electron density, even in datasets where the overall resolution is higher. The side chains in these regions are modelled to best fit the observed data, and refined using normal geometric restraints. Artificially fixing the side chains to match the more commonly observed rotamers would not accurately reflect the observed data.

Minor points:

1. Line 97: redefine the range of the flexible hinge as residues 728–734 (the loop region lacking β -structure), rather than 725–735.

The residues in the manuscript have been modified to cover residues 727-734. Overlay of D1 across multiple structures demonstrates that residue 727 is an initiating point for rotation of D2 relative to D1 via the flexible hinge region (as shown in SF 4), and as such is useful to consider as part of this region.

2. Line 146: revise "Supplementary Figure 1A" to "Supplementary Figure 1."

Thank you for spotting this typo - it has been corrected in the revised version (Line 149).

3. Line 178: revise "1.5A" to "1.5 Å."

Thank you for spotting this typo - it has been corrected in the revised version (Line 181).

4. Line 197: revise "WRN_GSK3" to "GSK_WRN3."

Thank you for spotting this typo - it has been corrected in the revised version (Line 201).

5. Lines 256–257: revise “in either the presence **of** absence of ssDNA” to “in either the presence or absence of ssDNA.”

Thank you for spotting this typo - it has been modified in the revised version.

6. Line 467: add a reference citation for Sommers et al.

Thank you for spotting this typo - it has been corrected in the revised version (Line 482).

Amanda Kennedy
CHARM Therapeutics,
B900, Babraham Research Campus
Cambridge, UK

1st January 2026

Dear Luciano Abriata,

Resubmission of manuscript COMMSBIO-25-7551-T

Thank you very much for sending our manuscript out to reviewers again and for your decision. We are delighted for the opportunity to submit a final revised version of the manuscript ‘*Structural insights into WRN helicase reveal conformational states and opportunities for MSI-H cancer drug discovery*’ for publication.

We have carefully addressed all comments raised and revised the manuscript and Supplementary Information accordingly. Below we address the two specific reviewer concerns and how they have been resolved in the revised version.

Reviewer 1 raised the important point that the structural basis for the observed variation in ssDNA binding was not stated clearly enough. In response, we revised the Discussion to explicitly clarify that in our WRN helicase-core constructs, ssDNA binding depends on coordinated engagement of the aromatic-rich loop (ARL; including F680 intercalation) together with the D2 ssDNA-interaction surface, and is therefore highly sensitive to changes in D1–D2 interdomain geometry across nucleotide- and inhibitor-bound states. We further strengthened this point by revising the final sentence of the relevant paragraph to link re-engagement of the ARL with ssDNA to restoration of the interdomain geometry required for coordinated ssDNA interactions and productive helicase activity (**Discussion, lines 309–312**).

Reviewer 3 suggested that potency metric definitions be included for clarity. We agreed that this would improve transparency and reproducibility, and therefore added definitions for **pIC50, pEC50, pglC50, and pDC50** in the footer of the new **Supplementary Table 1** (page 5 of the Supplementary Information), as requested.

We hope these revisions address the reviewers’ concerns and further strengthen the clarity and completeness of the manuscript for publication.

With best regards,

Dr Amanda Kennedy on behalf of all authors

Director, Translational Pharmacology

CHARM Therapeutics

amanda@charmtx.com | www.charmtx.com

Cambridge, UK